*Resource*

EMBO
Molecular Medicine

# Organotypic culture of human brain explants as a preclinical model for AI-driven antiviral studies

Emma Partiot [1,2,7], Barbara Gorda [1,2,7], Willy Lutz [1,2], Solène Lebrun[1,2], Pierre Khalfi [2,3], Stéphan Mora[2,3], Benoit Charlot[2,4], Karim Majzoub[2,3], Solange Desagher [1,2,3], Gowrishankar Ganesh[2,5], Sophie Colomb[2,6] & Raphael Gaudin [1,2 ✉]

## Abstract

Viral neuroinfections represent a major health burden for which the development of antivirals is needed. Antiviral compounds that target the consequences of a brain infection (symptomatic treatment) rather than the cause (direct-acting antivirals) constitute a promising mitigation strategy that requires to be investigated in relevant models. However, physiological surrogates mimicking an adult human cortex are lacking, limiting our understanding of the mechanisms associated with viro-induced neurological disorders. Here, we optimized the Organotypic culture of *Post-mortem* Adult human cortical Brain explants (OPAB) as a preclinical platform for Artificial Intelligence (AI)-driven antiviral studies. OPAB shows robust viability over weeks, well-preserved 3D cytoarchitecture, viral permissiveness, and spontaneous local field potential (LFP). Using LFP as a surrogate for neurohealth, we developed a machine learning framework to predict with high confidence the infection status of OPAB. As a proof-of-concept, we showed that antiviral-treated OPAB could partially restore LFP-based electrical activity of infected OPAB in a donor-dependent manner. Together, we propose OPAB as a physiologically relevant and versatile model to study neuroinfections and beyond, providing a platform for pre-clinical drug discovery.

**Keywords** Neurotropic Virus; Bunyavirus; Artificial Intelligence; Small Molecule; Tahyna Virus
**Subject Categories** Biotechnology & Synthetic Biology; Computational Biology; Microbiology, Virology & Host Pathogen Interaction

## Introduction

The study of the central nervous system (CNS) requires models that can reflect the physiological complexity of its structures to understand their functions. Cell culture in 2D gives access to molecular mechanistic properties of various cell types, but the cytoarchitectural organization of the tissue is not represented. During the last decade, technologies based on stem cell differentiation allowed the formation of cortical organoids recapitulating the layered cortical architecture of the brain. Although it represents an undeniable revolution, cortical organoids have two major flaws: they do not contain the full spectrum of cell types, and they mimic a fetal brain rather than an adult brain. Although much progress has been done in the field of brain organoids in the past couple of years, such as the introduction of microglia and blood vessels, the formation of multiregional assembloids, and electrical recording of local field potential (Ahn et al, 2021; Samudyata et al, 2022; Trujillo et al, 2019), it remains relatively primitive and a lot remains to be done before it can outperform animal models.

Rodent-based experiments are highly prevalent in studies investigating mechanisms underlying neurological perturbations. Beyond the ethical concerns associated to animal experimentation, humans have a far more complex brain functioning than mice, as quantitatively demonstrated previously (Hodge et al, 2019). Moreover, experimental manipulation of live animals also affects their psychological state and behavior, which can deeply impact the neurobiology of their CNS at the molecular and cellular levels. The same would hold true for most mammalian animal models.

This issue necessitates the development of new in vitro models that mimic the human brain's architecture with higher fidelity. A model system that meets these requirements is the ex vivo culture of human adult brain slices freshly sampled from brain surgical resection. Studies have shown that these slices can be maintained in culture ex vivo for several weeks, preserving cell diversity, tissue architecture, and electrical activity (Ravi et al, 2019). The main limitations of this model, however, are that brain donors are rare, the amount of resected material is small and limited to certain regions of the brain, and the donors all exhibit neurological disorders (mostly epilepsy or cancer) to justify surgery.

With the aim of offering a complementary alternative to in vivo experimentation and organoids, we propose to use brain explants from deceased donors as primary material for 3D cell culture. Indeed, *post-mortem* brain slices can be maintained in culture for an extended period of time, while preserving their cytoarchitectural

[1]CNRS, Institut de Recherche en Infectiologie de Montpellier (IRIM), 34293 Montpellier, France. [2]Univ Montpellier, 34090 Montpellier, France. [3]CNRS, Institut de Génétique Moléculaire de Montpellier (IGMM), 34293 Montpellier, France. [4]Institut d'Electronique et des Systèmes IES, CNRS, 860 Rue de St - Priest Bâtiment 5, 34090 Montpellier, France. [5]UM-CNRS Laboratoire d'Informatique de Robotique et de Microelectronique de Montpellier (LIRMM), 161, Rue Ada, 34090 Montpellier, France. [6]Équipe de droit pénal et sciences forensiques de Montpellier (EDPFM), Univ. Montpellier, Département de médecine légale, Pôle Urgences, Centre Hospitalo-Universitaire de Montpellier, 371 Avenue du Doyen Gaston Giraud, 34285 Montpellier, France. [7]These authors contributed equally: Emma Partiot, Barbara Gorda. ✉E-mail: raphael.gaudin@irim.cnrs.fr

organization and electrophysiological properties (Partiot et al, 2022; Qi et al, 2019; Schwarz et al, 2019). This model has the strong advantages to be of human adult origin, to provide large number of samples (a human brain is about 1.3 kg of primary biological sample), from any region of the brain, and at a high frequency. The cumulation of these characteristics make it a unique approach, unmatched by any other model, combining practical, physiological, and scalable primary material.

We chose here to evaluate the use of Organotypic culture of Post-mortem Adult human cortical Brain explants (OPAB) as a tool to study neuroinfection and antiviral strategies. To this end, we used the neurotropic orthobunyavirus Tahyna virus (TAHV) as a model. TAHV is an enveloped negative-strand RNA virus from the California serogroup (CSG). CSG orthobunyaviruses can induce neuropathogenesis in mice, and TAHV was shown to replicate in the brain of immunocompetent mice upon intraperitoneal injection(Evans et al, 2019). Furthermore, La Crosse virus (LACV), another CSG orthobunyavirus, is known to cause significant neurologic deficits in children (McJunkin et al, 2001), and a recent review associated CNS diseases to 27 orthobunyaviruses (Edridge and van der Hoek, 2020). Yet, the molecular determinants governing neurological disorders associated to orthobunyaviruses are largely unknown. Because orthobunyaviruses are tri-segmented and highly prevalent, it is to be expected that reassortments occur, leading to novel emerging viruses of potential health concern. Thus, it is important to study those viruses and test antiviral strategies in relevant models. We show that TAHV can replicate in OPAB and that this system can be used to evaluate the efficacy of the small molecule RG10b, that we previously described as an anti-coronavirus antiviral (Bakhache et al, 2021). Indeed, RG10b displays potent and persistent antiviral activity against TAHV in LUHMES-derived neurons and OPAB. Moreover, we found that TAHV perturbs the local field potential of OPAB, which allowed the development of a predictive tool based on artificial intelligence (AI) to evaluate the infectious status of OPAB. As a proof-of-concept, we show that RG10b can partially restore the electrical activity of OPAB back to a "non-infected" status, although the molecule may have additional effect that remain to be characterized.

# Results

## Pipeline for the culture of organotypic post-mortem adult brain slices (OPAB)

The preparation of OPAB is detailed in the method section and illustrated in Fig. 1A. Briefly, the frontal and parietal regions of the cortex were dissected and small cubes (about 0.5–1 cm³) were cut perpendicularly to the longitudinal axis of the gyrus. The cubes were transported from the hospital to the lab within 30 min, and then sliced into 300-μm-thick sections using a vibratome. All the samples displayed cortical layers and a piece of the white matter. The slices were cultured in 6-well transwells and the lower part of the transwell contained N2 medium, while the upper part of the transwell where the sections are deposited was left in the open air, allowing ex vivo culture at the air–liquid interface. This organotypic culture of post-mortem adult brain slices is thereafter referred to as OPAB.

## Macroscopic observation of OPAB

Our protocol allows for long-term culture of the brain slices ex vivo. Further quality assessment of the tissue samples over time by histological imaging right after harvest (0 DIV) or after culture for 2 or 12 days in vitro (DIV), indicated that numerous neurons (NeuN marker) remained, although less neurons were observed after slice culture (Figs. 1B,C and EV1A, B). This is expected as tissue processing likely includes cutting of axonal projections leading to a subset of neuronal death. Interestingly however, the histological morphology and integrity of the tissue remained quasi-identical between 2 DIV and 12 DIV, suggesting that the culturing procedure implemented did not significantly damage the tissue. Observations of the immunohistochemistry (IHC) staining of oligodendrocytes (Oligo2 marker) did not show marked differences between all the samples (Fig. EV1A,B). However, IHC staining of astrocytes using GFAP antibodies highlighted that right after cutting, astrocytes expressed more GFAP, indicative of an activated status, which disappeared in the following days (Fig. EV1B). This observation was supported by RT-qPCR analyses showing a trend toward less GFAP-coding mRNA over time post organotypic culture (Fig. EV2A). To investigate whether the GFAP decrease was not the consequence of the astrocytes' death, we measured the relative amount of mRNA coding for SOX9, a nuclear marker of astrocytes in the adult cortex (Sun et al, 2017), and found that it tended to increase over time, although it did not reach significance due to the low number of samples we were able to analyze (Fig. EV2B). By immunofluorescence, we quantified the number of SOX9-positive astrocytes at different days of OPAB, and found a significant increase after 8 days in culture compared to earlier timepoints (Fig. EV2C,D). These data suggest that the cutting procedure may trigger initial immune activation that drops shortly after ex vivo culture. Analysis of mRNA levels coding for the pro-inflammatory cytokine Interleukin-1 beta (IL1β) showed that it is expressed right after cutting, but quickly decreased over time of culture (except for one slice at 4 DIV; Fig. EV2E). Moreover, we could not detect the presence of leukocytes (CD45), T lymphocyte (CD3), nor monocyte/macrophage (CD163) infiltration at 0 DIV (Figs. 1D,E and EV2F), suggesting that the explants did not experience neuroinflammation prior to the patient's death.

## OPAB composition and viability

To further characterize the tissue architecture of OPAB, we used high-resolution 3D confocal imaging and stained for astrocytes, neurons, microglia, and blood vessels (Figs. 2A–D and EV3A–C). We observed that OPAB had preserved 3D architecture, exhibiting post-mitotic neurons with pyramidal shapes and aligned organization (Figs. 2A and EV3A–C). These images highlighted that OPAB are composed of major neural cell types, with cell nuclei of "healthy" morphology.

To gain further insights onto the viability of the cells composing OPAB, we used a LIVE/DEAD fluorescent reagent and demonstrated that numerous cells were viable (Fig. 2E). Indeed, our data indicates that cells closest to the cut edge (Fig. 2E, upper left) of the slice were dying (red), while most cells remained healthy (green) as we moved away from the border of the slice (Fig. 2E, toward the lower right). To test the quality of OPAB over time, we measured Lactate Dehydrogenase (LDH) in the supernatant, an indirect

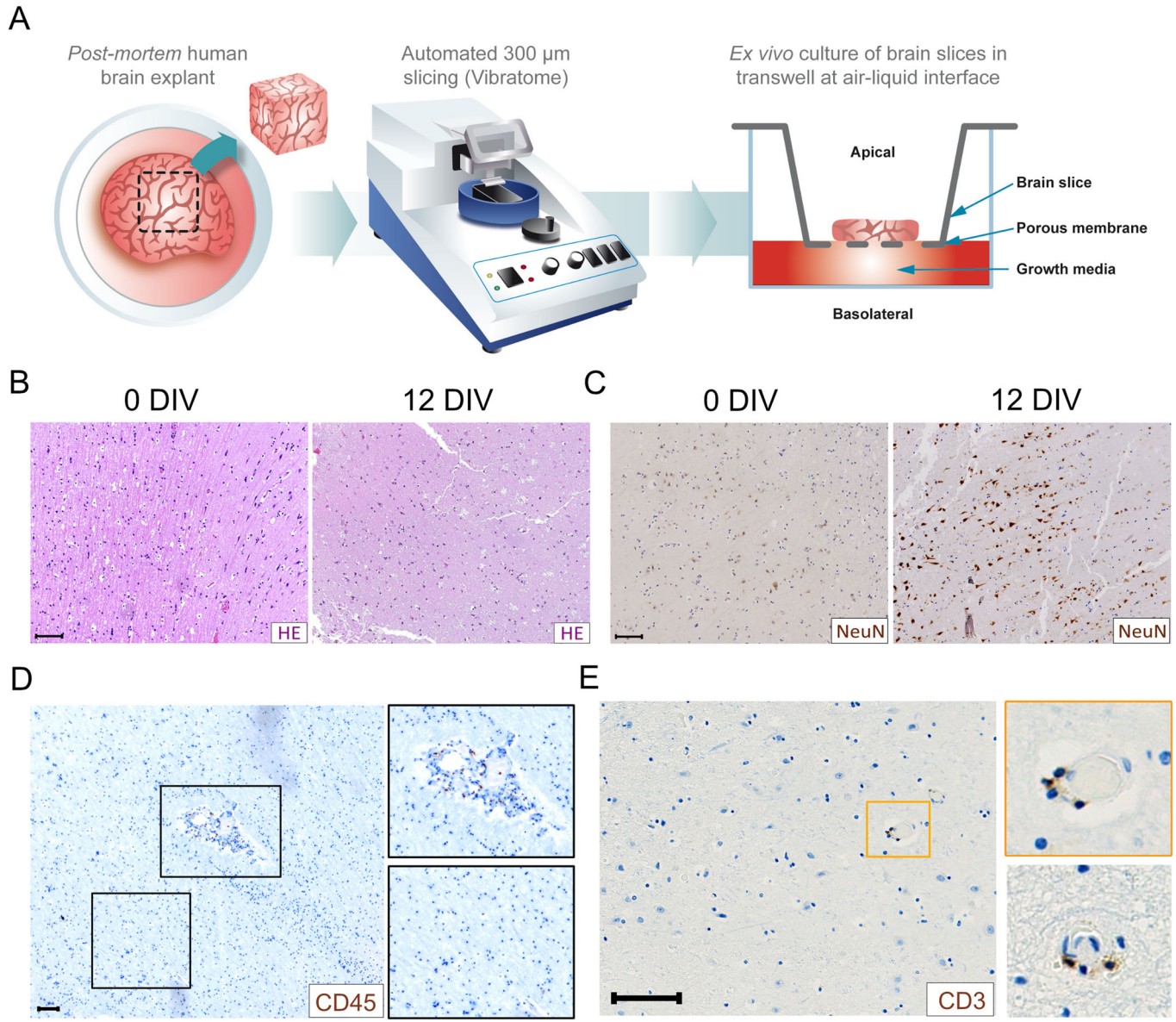

**Figure 1. Schematic procedure and histochemical characterization of organotypic human cortical slice culture (OPAB).**

(A) The scheme represents the procedure to obtain and culture OPAB. The first step of the procedure (left panel) is to harvest brain tissue by cutting small cubes (about 0.5 cm³) from the cortical area of interest (frontal or parietal). Then, 300 μm-thick tissue slices are cut using a vibratome (middle panel). The brain slices are transferred and cultured at air–liquid interface (right panel). (B, C) Immunohistochemistry of Hemalin-Eosin (HE; B) or NeuN (neuronal marker; C) of OPAB when harvested from patients (0 DIV), or after 12 days in culture (12 DIV). Scale bar: 100 μm. (D, E) Immunohistochemistry of OPAB at 0 DIV for leukocytes (D, CD45) and T lymphocyte (E, CD3). The right panels correspond to magnifications from the squares, and another zoom-in highlighting blood vessels with CD3-positive cells. Scale bar: 100 μm. Source data are available online for this figure.

reporter of cytotoxicity (Fig. 2F–H). Cytotoxicity was measured on the same slices over time at early timepoint (4 to 14 DIV, Fig. 2F) or late timepoint (50 to 60 DIV, Fig. 2G) post culture ex vivo and no significant differences of cytotoxicity were observed over time in both conditions. At the end of the culture, OPAB were lysed (positive control) and high LDH signal was measured, confirming that live cells were still present in the tissue. Moreover, we also incubated OPAB with Gambogic acid, a neurotoxic compound previously described (Malik et al, 2014), which increased LDH release in neurons (Fig. EV3D) and cleaved-capsase3 staining

intensity in OPAB (Fig. EV3E). Using OPAB from the same donors and at the same timepoints as in Fig. 2F, we showed that Gambogic acid increased cytotoxicity in OPAB over time (Fig. 2H), further showing biological reactivity of the OPAB and sensitivity of the LDH assay to account for neurotoxicity.

## Neuronal circuitry of OPAB

To investigate whether OPAB retained neuronal connectivity, we first imaged at high resolution, the synaptic components of OPAB.

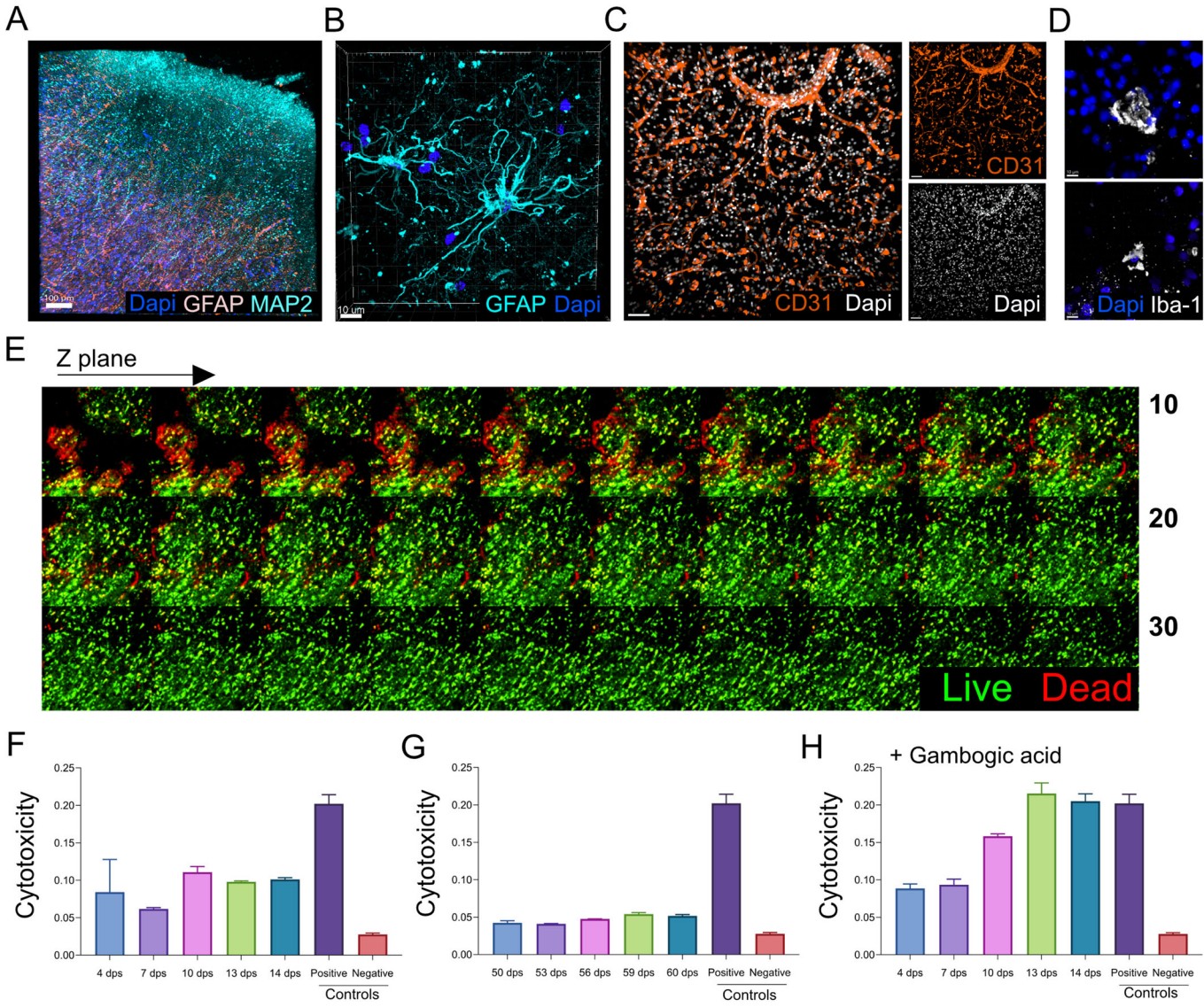

**Figure 2. Schematic procedure and histochemical characterization of organotypic human cortical slice culture (OPAB).**

(A) Three-dimensional confocal snapshot of a parietal OPAB cultured for 5 days ex vivo showing the organization of mature neurons (MAP2, cyan), astrocytes (GFAP, orange), and cell nuclei (Dapi, blue). Scale bar: 100 μm. (B–D) Snapshots from 3D confocal imaging (top view) of frontal OPAB, showing astrocytes (GFAP, **B**), blood vessel endothelial cells (CD31, **C**), and microglial cells (Iba1, **D**). Scale bars: 10 μm. (**E**) Frontal OPAB cultured for 16 days ex vivo stained with a Live/Dead marker, showing healthy cells (green), dead cells (red), and dying cells in yellow. The micrograph shows a gallery of confocal Z plans acquired at 1 μm interval, from the bottom of the slice near the coverslip (upper left) toward 30 μm within the slice (lower right). (**F–H**) The bar graphs correspond to the measurement of LDH release, accounting for cytotoxicity in untreated young (**F**) or older (**G**) OPAB, or young OPAB treated with 10 μg/ml Gambogic Acid (**H**), a neurotoxic agent, after indicated days in culture. Each graph corresponds to one donor and at least three individual slices were assessed per condition. Source data are available online for this figure.

Our observations of colocalized pre- and post-synaptic markers (Fig. 3A–C), revealed the presence of *bona fide* trans-synaptic structures along neurons. Hence, we further investigated OPAB electrical activity by measuring the spontaneous local field potential (LFP) of OPAB using 3D microelectrode arrays (3D-MEA (Heuschkel et al, 2002); Fig. 3D,E). These conical microelectrodes allow to penetrate up to 80 μm in the slice (which is 300–400 μm thick) and thus, to measure the electrical activity as close as possible to viable neurons, unlike planar electrodes. With this approach, we detected spontaneous LFP, suggesting that this model could be suitable for neural network studies.

Hence, we next assessed the inter-OPAB and inter-donor variability by plotting the absolute amplitude-frequency spectrum (Fig. 3F,G). We found that the curves from all slices had similar profiles, except for one (Fig. 3F, slice 7 from Donor B′ in green). This data indicates that interslice variability is relatively low or can be readily identified. Plotting the average values for each donor (excluding slice 7) showed an overall similar curve shape, although some donor specificity appeared (Fig. 3G). Of note, the conditions B and B′ correspond to slices from the same donor, for which LFP measurement was performed at 1-week interval. Among the four conditions, similar and specific inter-donor trends were seen by

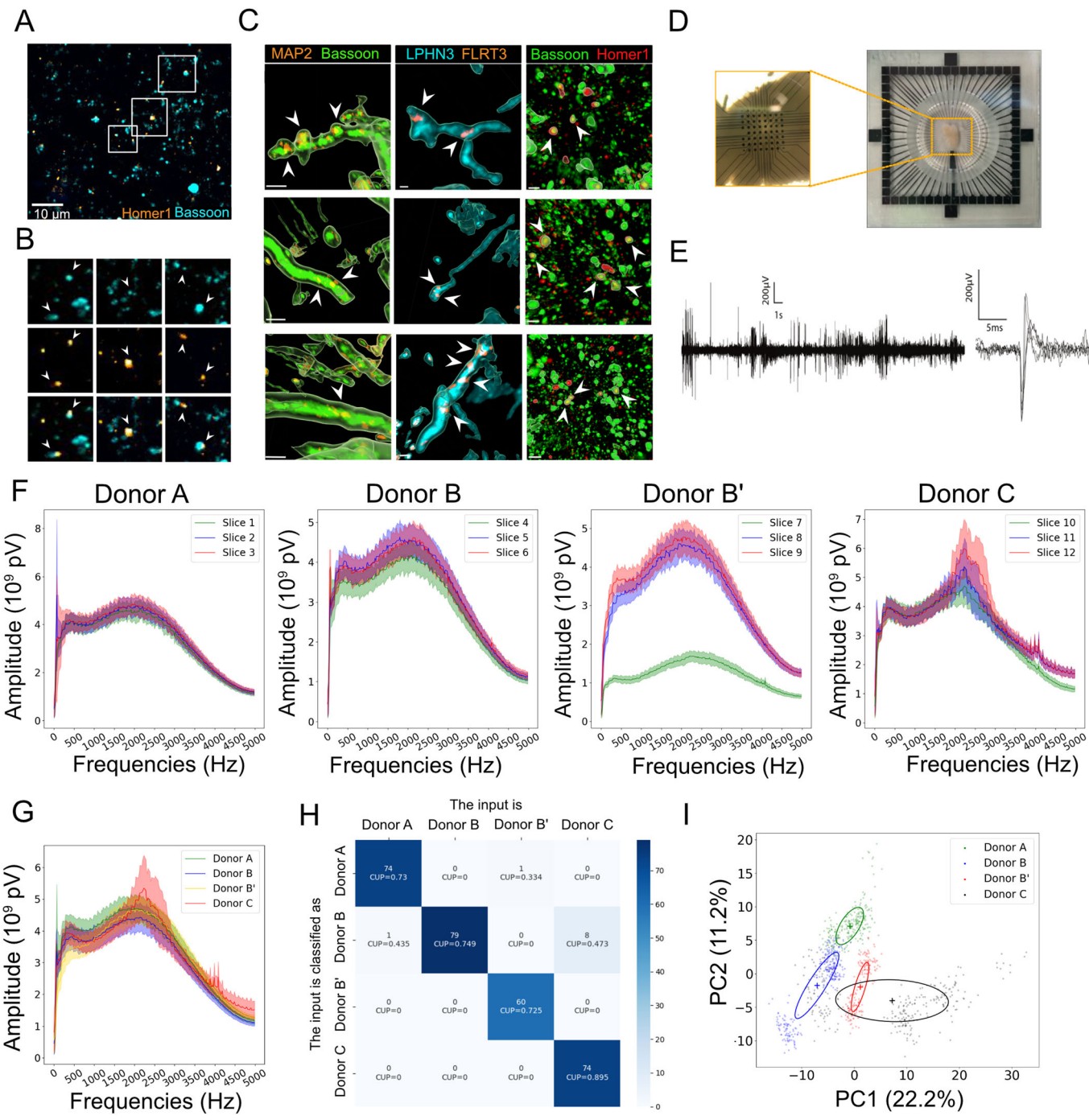

eye. Deeper analyses using a machine learning algorithm based on Random Forest Classifier (RFC) showed that each condition could readily be isolated from the other ones (Fig. 3H). The confusion matrix shows how much a sample from a particular donor can be discriminated from other donors solely based on its spontaneous LFP. A larger number in the matrix indicates higher similarity between the donors along the row and column and lower ability of the RFC to segregate them. As the work was performed with 4 donors, the confidence upon prediction (CUP; see Methods for details) is 1/4 = 0.25. Note the near chance non-diagonal blocks

indicating each donor sample was different from the others. Interestingly, donor B and B′ were also found to be different, although coming from the same donor (but LFP measurement was performed at 1-week interval). Principal component analysis of the four conditions further showed that each condition's LFP has its own cluster, barely overlapping with each other (Fig. 3I). In conclusion, the analysis of the electrical activity shows that OPAB exhibits specific LFP signatures and that donors shall be investigated separately to circumvent the heterogeneity of the responses.

**Figure 3.   Assessment of OPAB electrical activity.**

(A, B) The micrographs represent high-resolution images of frontal OPAB stained with the presynaptic marker Bassoon (cyan) and post-synaptic marker Homer-1 (orange). Scale bar: 10 μm. (B) Magnification of Bassoon-Homer-1 colocalizing events from the white squares in (A). (C) Three-dimensional confocal imaging of neurons (MAP2), presynaptic proteins (Bassoon, FLRT3) and post-synaptic proteins (Homer-1 and LPHN3) in frontal OPAB. White arrows highlight trans-synaptic structures, positives for pre- and post-synaptic markers. Scale bar: 2 μm. (D, E) Frontal OPAB were seeded onto 3D-MEA (D) and electrical activity was recorded. (E) Example of traces of extracellular electrical signal as a function of time, recorded from one electrode (left), and overlay cut-outs showing the shape, amplitude, signal to noise ratio, and dynamics of isolated spike signals. (F) Single-donor assessment of OPAB spontaneous electrical activity. Data were processed as per LFP data processing indications (see Methods for details). Data is displayed across slices for each donor and the mean +/− SD from 3 recordings per slice is shown. Conditions for Donor B and B' correspond to a single donor for which electrical activity recording was performed twice, at 1-week interval. (G) Frequency spectrum of the spontaneous LFP of OPAB displayed across slices across donors. Data shows mean +/− SD from 3 slice per condition. (H) Confusion matrix of the Random Forest Classifier (RFC)-based machine learning algorithm trained across donors, and tested on the same conditions ("unseen" dataset). In every case, the first number describes the number of inputs from the X axis classified as the condition in the Y axis. $CUP_{min} = 0.25$. (I) Principal component of OPAB LFP across donors. The confidence ellipses represent the covariance of the given variables principal component 1 (PC1) and principal component 2 (PC2). The percentages of the X and Y axis represent the variance ratio of the components. Source data are available online for this figure.

## Infection by TAHV and antiviral activity of RG10b and Rottlerin in Vero E6 and LUHMES-derived neurons

To evaluate the use of OPAB for the preclinical study of viral neuroinfections, we used the orthobunyavirus TAHV as a neurotropic infection model. First, we optimized TAHV infection in the permissive Vero E6 cell line to determine the standard levels of replication and inhibition in a well-controlled system. We tested 3 potential antiviral compounds: Bafilomycin A1 (BafA1), an inhibitor of the vATPase pump that prevents acidification of endosomes and thus, virus-host membrane fusion (Windhaber et al, 2022); Rottlerin, a recently described compound that inhibits La Crosse virus (LACV), another CSG orthobunyavirus closely related to TAHV (Ojha et al, 2021); and RG10b, an antiviral compound that we recently described as a potent inhibitor of hCoV-229E and SARS-CoV-2 infections (α- and β-coronaviruses, respectively) (Bakhache et al, 2021). All three molecules potently inhibited TAHV RNA replication in Vero E6 cells (Fig. 4A,B). We next evaluated the half-maximal inhibitory concentration ($IC_{50}$) of RG10b using plaque assay and RT-qPCR (Fig. 4C,D) and found $IC_{50}$ of 0.76 and 0.64 μM respectively, while low cytotoxicity was observed (<20%; Fig. 4D). In comparison, Rottlerin had an $IC_{50}$ measured at 1.33 μM and $CC_{50}$ of 11 μM against LACV in Vero cells (Ojha et al, 2021).

To study more specifically the anti-TAHV potency of RG10b in a neuronal context, we used the Lund Human Mesencephalic (LUHMES) neuronal cell line (Lotharius et al, 2005), which can be homogenously differentiated into post-mitotic neurons in one week (see (Scholz et al, 2011) and Methods for details). As we did not have access to commercial antibodies against TAHV, we optimized a single molecule FISH (smFISH) assay to detect TAHV RNA by confocal fluorescence microscopy. We confirmed in Vero E6 that this method was specific for infected conditions and exhibited marginal background staining in non-infected cells (Fig. EV4A,B). Next, we showed that TAHV readily replicated in the LUHMES-derived neurons as shown by the bright smFISH staining, which could not come solely from incoming particles (Fig. 4E). Quantitative evaluation of the effect of RG10b on TAHV infection of neurons was performed by RT-qPCR at different time post-infection. We observed that TAHV RNA levels were significantly increasing over time in the non-treated condition, while RG10b decreased TAHV replication by about 10- and 50-fold at 24 and 48 hpi respectively, and UV-inactivated TAHV was unable to replicate (Fig. 4F). Furthermore, neurotoxicity was observed upon TAHV

infection, and RG10b treatment was able to significantly prevent it, as reflected by LDH assay (Fig. 4G).

## Infection by TAHV and antiviral activity of RG10b and Rottlerin in OPAB

The potency of RG10b and Rottlerin was then evaluated in OPAB, to represent a more physiological context of infection. OPAB were infected with $10^6$ plaque forming units (PFU) of TAHV for 2, 4, or 7 days in the presence or absence of 20 μM of Rottlerin or 10 μM of RG10b. We found that TAHV replication starts at 2 dpi, and reaches a plateau by 4 dpi (Fig. 5A, cyan bars). We confirmed that the detected RNA levels were not coming from the inoculum by using a UV-inactivated TAHV, which was unable to replicate in OPAB (Fig. 5B). Both RG10b and Rottlerin seemed to decrease replication at 4 dpi, although it did not reach significance. At 7 dpi however, we found that only RG10b was able to significantly reduce TAHV replication in OPAB by about 5-fold, going back to background levels (Fig. 5A, orange bars). In comparison, treatment with Rottlerin (same procedure, higher concentration than for RG10b) was unable to inhibit TAHV replication at 7 dpi, as viral RNA levels were similar to DMSO-treated samples (Fig. 5A, gray bars). Cytotoxicity, as measured by LDH assay, was negligible in all conditions (Fig. 5C). Of note, no newly produced particles were detected in the supernatant of OPAB, even in the Mock-treated samples (Fig. 5D), which could be due to the virus release occurring within the slice, or through cell-to-cell transfer. Together, our data indicates that RG10b is a potent antiviral compound against TAHV neuroinfection and that OPAB represent a useful tool for preclinical therapeutic assessment.

## TAHV perturbs OPAB local field potential

To further characterize how TAHV infection impacts OPAB LFP, we used RFC-based machine learning analyses similar to Fig. 3H. Here, the algorithm was trained to discriminate the LFP of OPAB infected with TAHV for 48 hpi versus mock-infected controls. We confirmed that TAHV-exposed OPAB were readily discriminated from their mock counterparts with >99% accuracy at 48 hpi (Fig. 6A). As controls, we measured the LFP of the same slices at time 0- or 30-min post-infection and at those timepoints, the algorithm was unable to predict the infected/non-infected status (accuracy values around 0.5 indicates random classification), highlighting that the prediction was not based on OPAB-to-

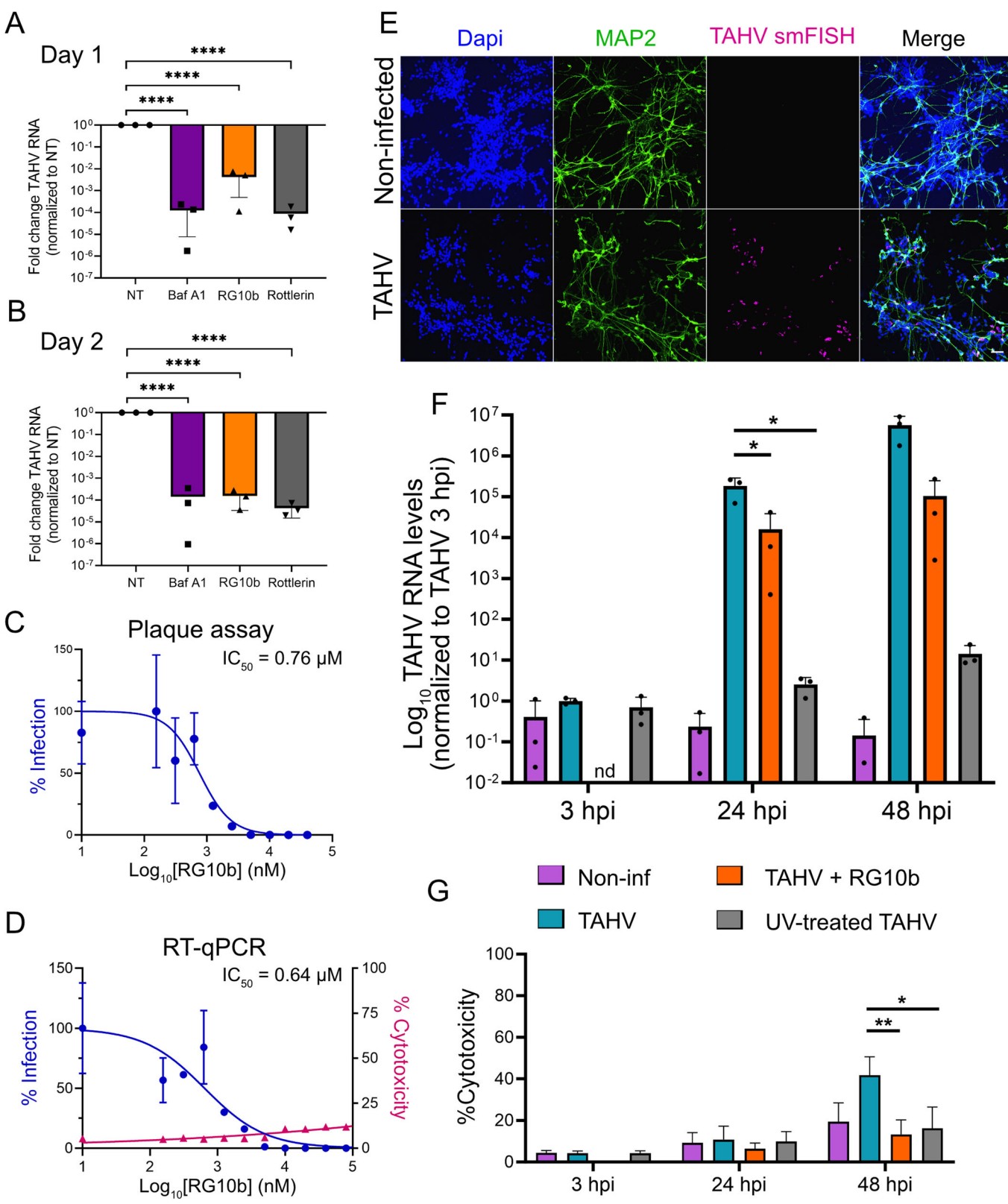

Figure 4.  TAHV infection and antivirals' evaluation in Vero E6 and neuronal cells.

(A, B) Vero E6 cells were infected for 1 (A) or 2 (B) days with TAHV at MOI 0.1 either non-treated (NT) or treated with 100 nM Bafilomycin A1 (Baf A1), 10 µM RG10b or 20 µM Rottlerin. Quantification of TAHV RNA, normalized by GAPDH RNA (control), was measured by RT-qPCR and data are represented as fold change to the TAHV infected non-treated (NT) condition. The data correspond to the mean $+/-$ SD from three individual experiments performed in duplicates. Unpaired t-test *p* value < 0.0001 (****). (C, D) RG10b $IC_{50}$ and $CC_{50}$ measurement in Vero E6 cells by RT-qPCR (C) and plaque assay (D). Data normalized to highest Ct values (RT-qPCR) and titer (Plaque assay) = 100% infection. Cytotoxicity measured by LDH assay and normalized to positive control = 100%. The data correspond to the mean $+/-$ SD from two individual experiments performed in duplicates. (E) Neuronal LUHMES cells were infected with TAHV for 48 h at MOI 0.1 and stained with anti-MAP2 antibody and smFISH targeting TAHV RNA. Confocal snapshots show *bona fide* neuronal differentiation (MAP2, green), and TAHV infection (TAHV smFISH, magenta). Scale bar: 50 µm. (F, G) Neuronal LUHMES cells were infected with TAHV for indicated time at MOI 0.1 and TAHV RNA expression levels were measured by RT-qPCR (F), and cytotoxicity was measured in the supernatant using LDH assay (G). The data are means $+/-$ SD from biological duplicates or triplicates from three independent experiments. Unpaired t-test *p* value < 0.05 (*), < 0.01 (**); nd: not determined. Source data are available online for this figure.

OPAB slice intrinsic heterogeneity. Moreover, at 96 hpi, our algorithm (trained with 48 hpi spontaneous LFP) was still able to predict accurately the infection status of OPAB (Fig. 6A), suggesting that TAHV profoundly and durably impacts the OPAB's electrical activity. Dimensional reduction plotting by principal component analysis (PCA) confirmed that Mock and TAHV infected samples exhibit distinct electrical features (Fig. 6B). Similarly to our observations in Fig. 3H, multi-donor analysis of the amplitude of the electrical signal as a function of its frequency did not allow to observe obvious differences between Mock and TAHV samples (Fig. 6C), further highlighting the need for AI-assisted analyses. Indeed, investigating which of the frequencies contributed the most for accurate Mock/TAHV discrimination clearly demonstrated that each donor exhibits a unique LFP signature to discriminate TAHV from Mock samples (Fig. 6D), reinforcing our observation that OPAB LFP should be individually examined, allowing interslice, but not inter-donor merging.

## RG10b partially reverts the TAHV-induced electrical perturbation in OPAB

To evaluate the capacity of RG10b not only to prevent TAHV replication in the brain (Fig. 5A), but also to restore "normal" brain function, we used our machine learning algorithm to evaluate whether RG10b-treated TAHV-infected OPAB could be misclassified as mock-infected (Fig. 6E). Strikingly, we found that RG10b treatment induced the partial misclassification of TAHV-infected OPAB LFP as "Mock" (Fig. 6E, see high values in the first row, third column for donor A and C). However, one donor did not respond to the RG10b treatment and the electrical signal remained associated to TAHV infection (Fig. 6E, first row, third column of donor B).

Taken together, our data shows that OPAB electrical activity measurement represents a suitable approach to study antiviral potency in a physiological system, and suggests that it could become a powerful tool for personalized medicine.

## Discussion

There is an urgent need for physiologically relevant in vitro models, as animal experimentations are increasingly questioned and should be kept to a minimum. Using human tissues coming from post-mortem brain explant represent a complementary strategy to current 3D models that must be considered for preclinical evaluations of drugs, but also to further our understanding of the

basic molecular mechanisms underlying physiopathological processes. Here, we propose to exploit OPAB as a complementary 3D model of cortical neural cells which can be widely implemented in research and development and preclinical studies, and we provide a proof-of-concept of its use to study viral neuroinfection and antiviral potency.

We show that OPAB are easy and relatively cheap to prepare, although one of the first barrier to implement this approach in labs is probably the legal authorizations and associated administrative burden. As for brain organoid research (Sawai et al, 2019), ethical considerations are needed to define best practice when using post-mortem explants in the lab. The organotypic culture of post-mortem human explants has been implemented recently to study SARS-CoV-2 infection of the lung and brain (Partiot et al, 2022; Schaller et al, 2021). An even better model to OPAB might come from the culture of human brain slices derived from neurosurgical resections (Schwarz et al, 2019), but such biological material is relatively rare and comes from patients with neurological disorders, potentially of genetic origin. The OPAB provides a complementary model, which can be obtained more frequently and in larger quantities. This approach has however the limitation to be highly heterogenous, because of the numerous parameters to be considered for each donor, including the post-mortem interval (PMI), the cause of death (CoD), co-morbidities and treatments. Our study provides a pipeline that takes into account the inter-donor variability and unbiasedly normalize it through AI-aided analyses, but we acknowledge that further analyses with a larger number of donors would be needed to correlate clinical metadata to ex vivo OPAB features.

Our data shows that OPAB are viable for months, although the initial slicing could injure the explant. This limitation stems from the intrinsic property of this approach, but we found that keeping the slices in culture for a few days allows initial astrocyte activation response to attenuate. Previous reports have observed some neuroinflammation upon brain slice culture of microglia that subsides after two weeks (Delbridge et al, 2020). Because of the post-mortem nature of the OPAB and the traumatic slicing procedure, such neuroinflammation is to be expected. Interestingly, immediately after resection, OPAB exhibited marked astrogliosis (high GFAP staining), but no leukocyte infiltration. However, observation of the OPAB at 12 DIV showed a marked decrease of GFAP signal, suggesting that the OPAB restore a non-inflammatory phenotype ex vivo. Our culture method also seems appropriate as no further cell death nor cytotoxicity could be detected at later timepoints in culture. Together, we show that OPAB comes with some limitations that we document here, but

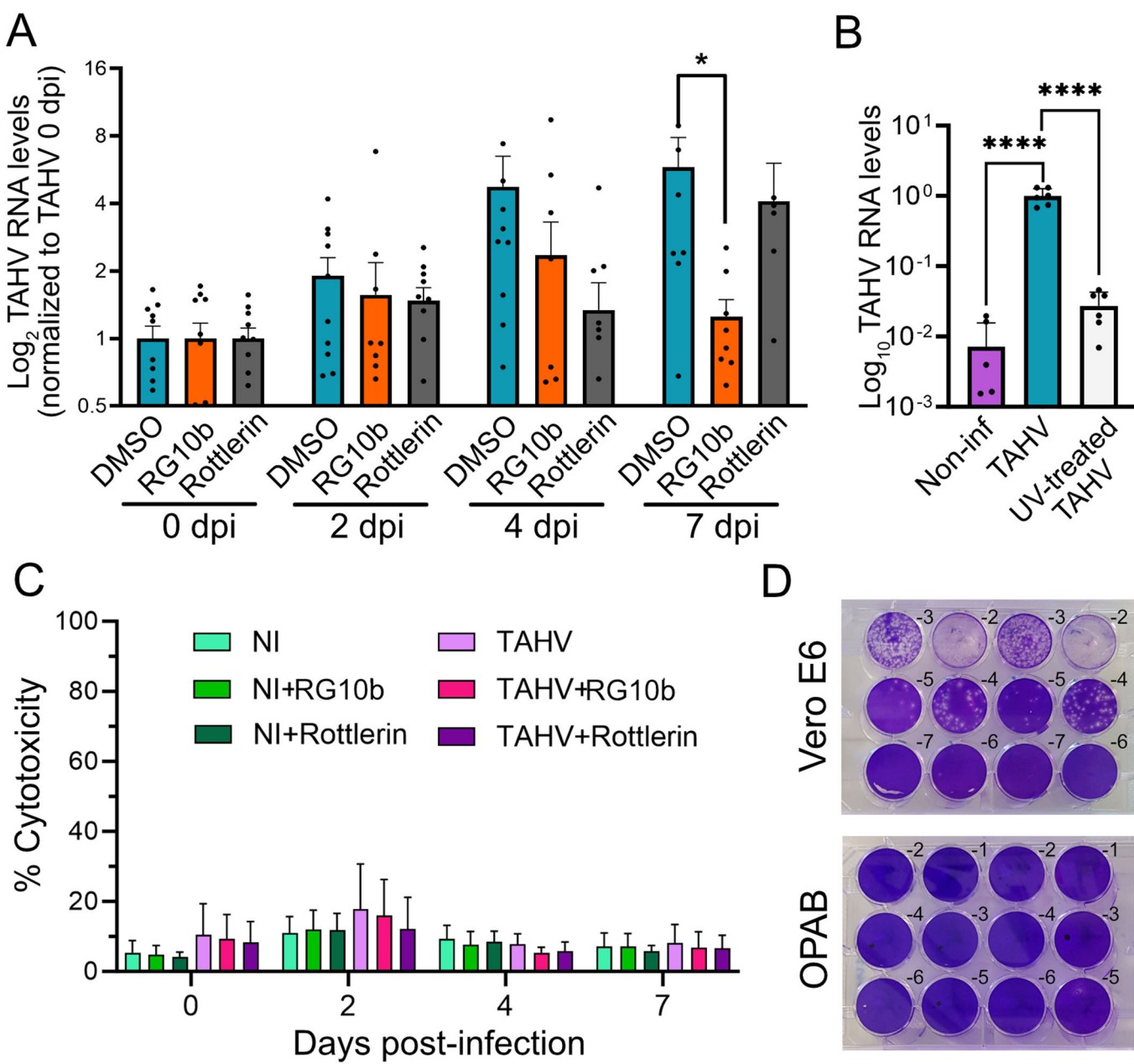

**Figure 5. TAHV infection and antivirals' evaluation in OPAB.**

(A) Parietal OPAB from 5 donors infected with $10^6$ pfu TAHV for indicated days were either DMSO-treated (DMSO) or treated with 10 µM RG10b or 20 µM Rottlerin. Quantification of TAHV RNA, normalized to GAPDH mRNA levels, was measured by RT-qPCR and data are represented as fold change normalized to conditions at 0 dpi to equal 1. The data correspond to the mean $+/-$ SEM from 5 donors ($n = 10$ slices for 0, 2, and 4 dpi and $n = 8$ slices for 7 dpi) performed in duplicates/triplicates. Unpaired t-test $p$ value $= 0.031$ (*). Differences between all other conditions are non-significant. (B) OPAB were either non-infected (Non-inf), infected with $10^6$ pfu TAHV or with equivalent volume of UV-inactivated TAHV. After 48 hpi, quantification of TAHV RNA, normalized by GAPDH RNA, was measured by RT-qPCR. The data correspond to the mean $+/-$ SD from 5 or 6 slices from 2 donors. Unpaired t-test $p$ value $< 0.0001$ (****). (C) Cytotoxicity in non-infected (NI) and TAHV-infected OPAB with 10 µM RG10b or 20 µM Rottlerin treatment at 0, 2, 4, and 7 dpi. Cytotoxicity was determined by measuring levels of released LDH in the culture supernatant and normalized to a positive control (lysed OPAB) $= 100\%$. Data are mean $+/-$ SD and with $n = 4$ or 5 slices from at least 2 donors. No significant differences were measured. (D) Example of plaque assays from TAHV production in Vero E6 cells at 2 dpi and from harvested supernatant from TAHV-infected OPAB at 7 dpi. Source data are available online for this figure.

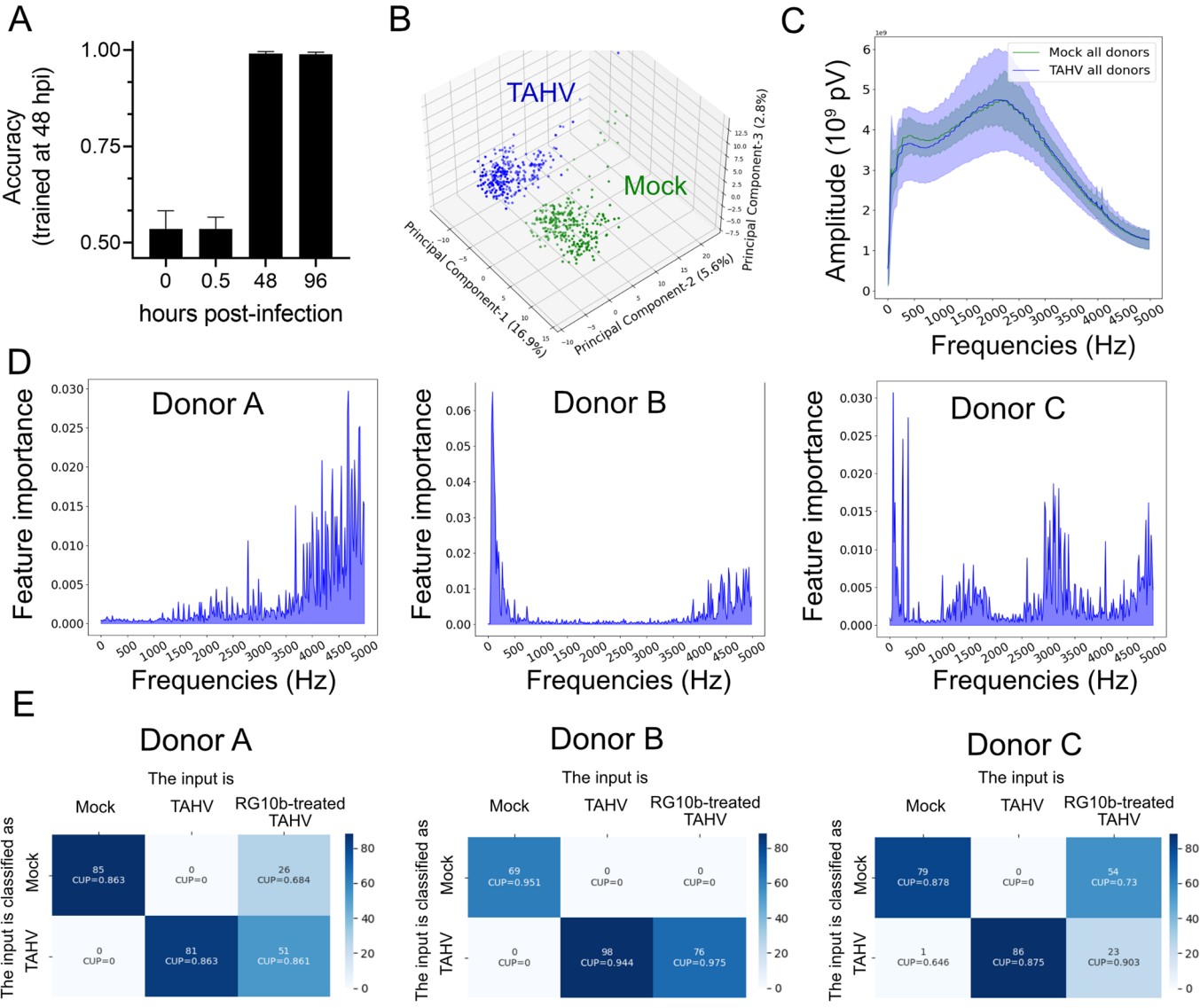

**Figure 6. Machine learning algorithm to evaluate OPAB neurohealth based on LFP.**

(A) Accuracy of RFC-based machine learning algorithm trained on the Mock-TAHV classification of LFP recording at 48 hpi, tested on the same OPAB at 0, 0.5, 48, and 96 hpi. The model shows high efficiency when tested on 48 and 96 hpi LFP recordings, and presents quasi-random predictions when tested on 0 and 0.5 hpi LFP recordings. Data are mean $+/-$ SD from 3 donors, 3 samples per donor, 3 recordings per samples. (B) The dot plot represents the first three Principal components after a principle component analysis (PCA) of the amplitude-frequency spectrum for the Mock and TAHV conditions recorded at 48 hpi. (C) Frequency spectrum for Mock and TAHV infected conditions recorded at 48 hpi, across all donors. The X-axis are the 300 values used to train the RFC model, and the unit is reported to the equivalent frequencies. The standard deviation is obtained across all slices of all donors. The outlier OPAB slice 7 shown in Fig. 3F donor B' has not been included in the analysis. (D) The graphs show Features importance for classification between Mock and TAHV-infected conditions recorded at 48 hpi, across donors. The X axis are the 300 features used to train the model, and the importance is scaled and quantified by arbitrary units. For each donor, the feature importance profiles are strongly different, showing that the prominent frequencies affected were different in each donor. (E) Confusion matrix of RFC model trained on condition Mock and TAHV-infected recordings at 48 hpi, and tested on those conditions in the presence of 10 µM RG10b (RG10b-treated TAHV) and recorded at 48 hpi. The first number describes the number of inputs from the X axis classified as the condition in the Y axis. For the donors A and B, RG10b-treated TAHV is partially classified as Mock and as non-treated TAHV. $CUP_{min} = 0.5$. Source data are available online for this figure.

this model has numerous advantages that make it a very valuable approach to model neurological diseases. Whether this model is suitable to investigate synaptic plasticity and to address our understanding of basic neurocognitive functions in vitro, remains to be established.

Antiviral research requires the use of more physiological models to evaluate potency. Indeed, it is becoming clear that high-

throughput screenings in monolayer cell lines yield very large number of hits that prove to be ineffective in more complex 3D cultures, primary cells, and animal models (see for instance (Koban et al, 2020)). Here, OPAB represents a powerful approach to assess the potency of antiviral compounds against neurotropic viruses, as it is made of different types of primary cells that maintain their tri-dimensional organization. Orthobunyavirus is a family of viruses

for which no antiviral drugs nor vaccine are available. Moreover, although the health burden of TAHV and other orthobunyaviruses is not massive, their prevalence worldwide is exceptional (Edridge and van der Hoek, 2020), and one should not wait for a pandemic to start worrying about these viruses. Orthobunyaviruses are tri-segmented, with potential for genome reassortment and new virus emergence, and because they infect a wide spectrum of host, they represent a non-negligible health threat. Previous studies reported on the use of organotypic brain slices from animal models (Ayala-Nunez et al, 2019; Ferren et al, 2021; Grabowski et al, 2018), but antivirals with specific human host targets may not be very potent in these systems. An elegant study recently used human brain slices from brain surgery to study Oropouche virus (OROV) infection, allowing to study virus tropism and neurological perturbations in greater details (Almeida et al, 2021). In our study, we did not have suitable antibodies that recognize TAHV by immunofluorescence and thus, could not investigate the tropism of TAHV, but previous study suggested that the main target of TAHV in the brain are likely neurons (Evans et al, 2019). Here, we showed that TAHV successfully infected neuronal LUMHES cells, through the development of TAHV-specific smFISH. This approach could not yet be optimized in OPAB, as thick sample preparation differs from monolayer of cells. Nevertheless, the sensitivity and specificity of smFISH makes it a technique of choice for future investigations.

As neurons are the primary vectors of electrical transmission in the cortex, we took advantage of OPAB to evaluate the impact of TAHV on LFP by implementing a custom-made machine learning algorithm. Artificial intelligence (AI) is increasingly used to help for decision-making in cancer diagnostics (Huang et al, 2020), and biologists start to harness the power of AI to obtain mechanistic insights and disease modeling (Greener et al, 2022). In particular, machine learning is getting popular in chemoinformatics for lead compound optimization by enabling the prediction of antiviral compounds with desired properties (Amendola and Cosconati, 2021; Gawriljuk et al, 2021). AI-based analysis was also applied in cell virology to discriminate between non-infected cells and infected ones, clearly demonstrating the feasibility of using AI for accurate prediction (Andriasyan et al, 2021). While the infection status of a cell may be monitored by various means, "neurohealth" is a convolved concept that requires integrated model systems. Here, we chose to measure the electrical signals of OPAB as a surrogate to physiological brain function. Indeed, this approach has the advantage to focus on physiological neuronal circuitry as a whole rather than an infected/non-infected status. Antivirals targeting viral replication are obviously of high interest, but innovative strategies to treat the symptoms associated to the exposure of the brain to viral infections shall be sought for. This approach may reflect on the level of OPAB physiological state ("healthiness"), comparable to the analysis of a micro-electroencephalogram (micro-EEG). Here, we developed an RFC-based algorithm able to predict with high accuracy whether OPAB were exposed or not to TAHV, solely based on their LFP signature, indicating that TAHV impacts physiologic brain slice LFP. RFC was chosen because it provided most efficient LFP discrimination between samples compared to Support Vector Machine (SVM) and Stochastic Gradient Descent Regressor (SGDR) during the early stages of the algorithm's development. RFC has also the advantage to return feature weights to pinpoint the contribution of each frequency in prediction outcome. Although other types of machine

learning strategies may also perform well on LFP analyses, this work highlights the promise of how electrical activity may be unbiasedly analyzed using AI.

As a proof-of-concept, we used RG10b, an antiviral molecule under development in the lab, to evaluate its potential to rescue TAHV-induced LFP perturbations. RG10b is a phenoxy-acetyl-thioureido-benzenesulfonamide-based molecule that showed potent antiviral activity against several coronaviruses (Bakhache et al, 2021). We had no a priori when testing RG10b against orthobunyaviruses, as the replication of these viruses greatly differs from that of coronaviruses, but as this molecule has broad-spectrum potential, impairing intracellular dynamics, we used it as a model to test our system. We show that RG10b is potent against TAHV in Vero cells, LUHMES-derived neurons and OPAB, while exhibiting low cytotoxicity. Strikingly, RG10b could partially restore a non-infected LFP status for two out of three donors. This data is promising and further investigations will be needed to improve this component and fully characterize its mode of action. In the near future, structure-activity relationship studies in OPAB will be undertaken to obtain lead compounds of higher potency, as it may represent promising broad-spectrum antiviral agents, which could be repurposed in the case of future viral pandemics.

Our work combines the analyses of an antiviral compound exhibiting good antiviral potency in cell lines, retaining prolonged antiviral activity in our 3D model of OPAB, and partially restoring normal neuronal electrical activity. Hence, we propose that OPAB can be used for antiviral potency evaluation as well as for functional investigations.

# Methods

## Study approval

A research protocol approval has been obtained from the institutional review board (IRB from the CHU Montpellier, approval ID: 202000643) and the "Agence de Biomédecine" (agreement number PSF20-025), allowing human brain resection during autopsy for research purpose related to this application. The local prosecutors allowed the use of human tissue samples obtained from commissioned autopsies. Informed consent was obtained from the families of the deceased human subjects. The experiments conformed to the principles set out in the WMA Declaration of Helsinki and the Department of Health and Human Services Belmont Report.

## Donor enrolment criteria

To include a patient into our protocol of research, the post-mortem interval (PMI) was particularly important, choosing only corpses with a PMI between 2 and 12 h (Table EV1). Moreover, the cause of death (CoD) was also chosen to be the least damaging for the brain. Deaths with involvement of the cephalic extremity were excluded (e.g., car crash, gun shot, stroke…). Relatively young patients (<60 yo) and absence of corticosteroid treatment were preferred (except for patient P7). As far as possible, patients with severe neurological pathologies (epilepsy, neurodegenerative diseases, encephalitis, etc.) were excluded from the study. Xenobiotics intakes (treatment and toxics) and co-morbidities were documented, but the information was not always

available prior to OPAB preparation. SARS-CoV-2 viremia was also routinely tested and returned negative for all patients.

## Preparation and culture of OPAB

During autopsy, after incising and reclining the scalp, the skull was opened with an electric saw and the whole brain was extracted, preserving aseptic conditions as much as possible. After macroscopic examination of the brain to verify the absence of lesions, and having removed the cerebellum and brainstem, the frontal and parietal regions of the cortex were dissected by cutting several 2–3 cm-thick coronal slices of the brain. For histological analyses before organotypic culture, a thin longitudinal slice of each cube was immediately placed in cassettes and fixed in formalin. The remaining explants were cut in small cubes (about 0.5–1 cm³) perpendicularly to the longitudinal axis of the gyrus and kept in N2 medium at 4 °C before slicing.

The cubes were then transported from the hospital to the lab within 30 min at 4 °C, immersed in Neurobasal medium (Thermo-Fisher) supplemented with N-2 supplement (ThermoFisher), Glutamax (ThermoFisher) and Penicillin/Streptomycin (Thermo-Fisher). This medium is referred to as N2 medium. The cubes were embedded in 3% low melting point agarose (ThermoFisher) under sterile conditions and sliced in 300 μm thick sections using an PELCO Easislicer (Ted Pella). All the slices contained cortical layers and some white matter. The brain slices were transferred without the agarose onto cell culture inserts with 0.4 μm pore PET membranes (Millicell, Millipore) hanging in 6-well plates containing 2.5 ml of N2 medium in the lower chamber, while the upper part of the transwell where the sections are deposited was left in the open air, allowing ex vivo culture at the air–liquid interface (Fig. 1). The slices were cultured at 37 °C, 5% $CO_2$ and 95% humidity and medium was changed twice a week.

## Antibodies and reagents

The following antibodies were used for immunofluorescence in this study: rabbit polyclonal anti-MAP2 (1/100, GeneTex, #GTX133109), rat anti-CTIP2 (1/100, BioLegend, clone [25B6]), goat anti-GFAP (1/100, Novus Biologicals, #NB100-53809), mouse anti-Tuj1 (1/100, Genetex, #GTX631836), rabbit anti-Iba1 (1/100, Genetex, #GTX101495), mouse anti-CD31 (1/100, PECAM, Miltenyi Biotec, #130-108-038), mouse anti-Bassoon (1/50, Abcam, clone [SAP7F407]), Rabbit anti-Homer1 (1/50, Synaptic system, #160 003), Rabbit anti-LPHN3 (1/100, ThermoFisher, #PA533908), Goat anti-FLRT3 (1/50, R&D Systems, #AF2795), rabbit anti-Cleaved Caspase-3 (1/100, Cell Signaling Technology, clone [5A1E]), goat anti-SOX9 (1/50, R&D Systems, AF3075). The following antibodies were used for histology examination: Mouse anti-NeuN (1/400, clone A60, #MAB377, Millipore), rabbit anti-Oligo2 (1/100, clone EP112, #BSB2562, BioSB/diagomics), Mouse anti-GFAP (1/400, clone 6F2, #MO761, DAKO), Rabbit anti-CD3 (#05278422001, Roche), mouse anti-CD68 (clone KP1, #M0814, DAKO). The following dyes were used in this study: Dapi Nuclear Counterstain (Pierce), Hematoxylin and eosin (H&E), Live/Dead Viability/Cytotoxicity (ThermoFisher Scientific). Culture media and TrypLE were from ThermoFisher Scientific. Recombinant human basic Fibroblast Growth Factor (bFGF) and Glial cell line-Derived Neurotrophic Factor (GDNF) were from R&D systems.

Poly-L-ornithine, fibronectin, laminin, dibutyryl cAMP (dbcAMP) were from Sigma-Aldrich.

Bafilomycin A1 (BafA1) was purchased from Santa Cruz Biotechnology and diluted to 100 μM solution in DMSO. RG10b was synthesized by AGV Discovery and diluted to 10 mM solution in DMSO. Rottlerin was purchased from MedChemExpress and diluted to 20 mM solution in DMSO.

## Immunohistochemistry

Tissue explants were fixed in formalin and embedded in paraffin. Staining of 5-μm sections was performed with hematoxylin and eosin. Immunohistochemistry (IHC) was performed with Auto-stainer BenchMark ULTRA (Ventana Medical Systems), according to manufacturer's instructions. Briefly, paraffin sections were incubated at 60 °C for 16 min and then at 72 °C for dewaxing and at 95 °C in retrieval antigen buffer. Incubation for 30 min with antibodies was performed and revealed with DAB. Samples were washed and incubated with Hematoxylin II (counterstaining) for 12 min and with bluing reagent (counterstaining) for 4 min. Images were acquired on Leica Thunder equipped with a 3072 × 2048 K3C colored camera (Leica). Mosaic acquisitions were done with a 20×, NA 0.8 air Leica objective.

## Immunostaining and confocal imaging

OPAB were washed with phosphate-buffered saline (PBS) and transferred to flat bottom 24-well plate. Samples were fixed in 4% paraformaldehyde (PFA) for 1 h at room temperature (RT) and permeabilized for 24 h on a rocker with permeabilization buffer (0.5% BSA, 1% Triton in PBS). All subsequent steps were performed in permeabilization buffer with overnight incubation. Samples were labeled with primary antibody, washed, and labeled with appropriate secondary antibodies. After washes, the samples were incubated with RapiClear 1.52 reagent (Sunjin Lab) overnight at RT.

Image acquisition was performed on a spinning-disk confocal microscope (Dragonfly, Oxford Instruments) equipped with an ultrasensitive 1024 × 1024 EMCCD camera (iXon Life 888, Andor) and four laser lines (405, 488, 561, and 637 nm). A 20×, NA 0.8 air objective (Nikon) was used for whole sample imaging and a 60×, NA 1.4 (Nikon) oil-immersion long-distance objective was used for in-depth imaging of selected areas. Images were processed using FiJi (ImageJ) and Bitplane Imaris x64 (Oxford Instruments) version 9.2 and 9.7.

## RNA staining using smFISH

The smFISH procedure relies on the original methods described as smiFISH in Tsanov et al, 2016. Briefly, cells on #1.5 coverslips were fixed in 4% PFA and blocked with 0.5% BSA, 0.1% Triton in 1X PBS for 30 min at RT, then washed in smFISH wash buffer (2X SSC, 10% Formamide in ultrapure $H_2O$). A total of 680 ng of primary probes (purchased from Integrated DNA Technologies) targeting the three segments of the TAHV genome (see Dataset EV1) were pre-hybridized with 0.7 μl of 100 mM secondary Cy3 Y-Flap probe, 1 μl of NEB 3 buffer and filled up to 10 μl total volume with $H_2O$ and incubated for 3 min at 85 °C, 3 min at 65 °C, and 5 min at 25 °C. Slides were incubated with hybridization buffer and optional primary antibody

overnight at 37 °C protected from light for all the following steps. The coverslips were washed twice with pre-warmed smFISH wash buffer at 37 °C for 15 min and subsequently incubated in smFISH wash buffer supplemented with 0.5% BSA, 1 μg/ml DAPI and secondary antibodies for 1 h at 37 °C. Three washes in 2X SSC, 0.1% Tween in $H_2O$ were performed for 15 min at RT, prior to mounting of the coverslips on glass slides using Mowiol (Sigma-Aldrich).

| Hybridization buffer (100 μl per slide). | |
| --- | --- |
| Formamide 100% | 10 μl |
| 20X SSC | 10 μl |
| Dextran sulfate 40% | 20 μl |
| tRNA (stock 20 mg/mL) | 1,7 μl |
| smiFISH probe | 2 μl |
| VRC (stock 200 mM) | 1 μl |
| BSA (stock 20 mg/mL) | 5 μl |
| $H_2O$ | Up to 100 μL (50.3 μl) |

## Cytotoxicity assay

Cytotoxicity assay was done with CyQUANT LDH Cytotoxicity Assay (Thermo) and the experiments were performed according to the manufacturer's instructions. Briefly, supernatant from OPAB was harvested and incubated in 96-well plates with the LDH substrate for 30 min at RT. The stop solution was added to stop the reaction and the absorbance was read with a microplate reader (TECAN Infinite M Plex).

Live/dead Viability/Cytotoxicity Kit from Invitrogen was performed according to the manufacturer's instructions. Briefly, cells were incubated with the reagent for 1 h at 37 °C. The OPAB were washed 3 times with PBS, fixed with 4% PFA and treated with RapiClear over night at 4 °C. Images acquisition was done with spinning-disk confocal microscope (Dragonfly, Oxford Instruments) with the 20× NA 0.8 air objective (Nikon).

## Procedure for OPAB LFP recording

OPABs were placed on 3D microelectrode arrays (3D-MEA, 60 electrodes, Multichannel systems) with 100–150 μl of N2 medium at least 2–3 h before recording. Just before recording, the medium was removed and 100 μl of fresh N2 medium was placed on the OPAB in order to retain part of the slice at the air–liquid interface. Recording of mock-infected or infected OPAB, was performed for indicated timepoints with $10^6$ PFU TAHV and in the presence or absence of 10 μM RG10b. Three recordings of 1 min each were taken at 0, 0.5, 48, and 96 hpi per OBAP using the MEA2100-Mini-System with the Multi Channel Experimenter software (Multichannel system). The N2 medium with or without RG10b was replaced every day.

## Available data for machine learning and computational analysis

The electrical signal sampled at a 10 KHz frequency was recorded from 3D-MEAs composed of 60 electrodes (see above). Electrical signal was recorded at 0, 0.5, 48, and 96 hpi, with the mock status (referred as Mock), the TAHV infected status (referred as TAHV), the Tahyna virus infected RG10b-treated status (referred as RG10b-treated TAHV). We tested three donors (A, B, and C), donor B being tested twice (B and B′) at 1-week interval. For each condition, we observed three OPAB slices, and for each slice we performed three 1-min recordings.

## LFP data processing

Considering that OPABs do not cover the whole electrode array, to avoid recording non-covered area, we selected (for each recording) the 35 electrodes that presented the largest standard deviation across the whole recording, thus considered the most active. Once selected, each recording was cut into 30 pieces of 2 s recordings. Each 2-s recording was used as an entry in our dataset. Subsequently, for each 2-s sample, the 35 electrodes were analyzed. Each sample was first high-pass filtered over 50 Hz with a Butter filter of order 3. The first filter is a high-pass with cutout frequency of 50 Hz. Then, the odd harmonics of 50 Hz were filtered up to the 30th harmonic (equivalent to 50 Hz, 150 Hz, 250 Hz… up to 2950 Hz), because it showed non-specific spikes. Of note, the 50 Hz harmonics (odd and even) were not significant discrimination factors for classification. Each recording was then processed by a Fast Fourier Transform to obtain frequency data. The frequency data were averaged across all the selected electrodes of the same 2-s recording. Each 1-min recording thus gave 30 recordings, and each donor thus gave 270 recordings per condition.

Next, the dataset for subsequent training of machine learning algorithms and for miscellaneous analysis was generated. Because of the high sampling frequency, the number of frequency features were reduced to train the model in order to optimize the quality-computation ratio. To this end, each one of these recordings was down-sampled in frequency space to give 300 samples each of 16.6 Hz width.

## Machine learning algorithm for LFP analysis

To analyze the processed data, a Random Forest Classifier (RFC) algorithm was implemented. The whole dataset was separated into a training dataset containing 70% of the data, and a testing part containing the remaining 30%, which served as unseen data to test the model. The RFC algorithm uses the default parameter of the scikit-learn python library RandomForestClassifier Class, with the exception of the number of estimator (trees in the forest), which was set to 1000.

## Machine learning-based analysis

Given an untrained RFC model and the available data discussed in the part 'available data for machine learning analysis', our models were trained on different combinations of conditions. To test if the model is efficient, we trained the model on a certain set of conditions, then tested it on the same conditions, but with unseen data (testing dataset). If the model classifies correctly the unseen data, then it is considered efficient. We also used a model trained on certain conditions, and tested our model on a different set of conditions, to see how the model classified the data in this condition. This can be used to see if

conditions have certain similarities so that the model confuses them. This process can be summarized in a confusion matrix. This procedure acts as a control on the efficiency of the model, allowing to add other conditions to test.

After model training, the feature importance is computed by the model while training. It represents the relative weight each feature has when deciding, based on the Gini importance (or decrease in mean impurity). In other words, the larger the feature weight is, the earlier it helped the tree to predict an output.

## Principal component analysis

In parallel to the RFC analysis, a dimensions reduction analysis of the data was performed using principal component analysis (PCA). Our PCA algorithm used the default hyper parameters given in the scikit-learn python library version 1.2.1. The PCA algorithm was fit on a certain set of conditions (fitting set) to transform those conditions in an optimal way for the decomposition, then applied this decomposition to another set of conditions (testing set) to visually represent similarities between the training and testing sets. The explained variance ratio in PCA shows the variance in the data attributed to each of the principal components.

## Metrics for machine learning and computer science-based analysis

### Confidence upon prediction (CUP)

The efficiency of our RFC models can be quantified using the accuracy of the model, and its confidence upon prediction (CUP). Given a dataset of $n$ entries (processed electrical recordings), $n/2$ being from class A (for example 'mock'), and $n/2$ from class B (for example 'infected'), as follows:

| Entries | Class |
|---|---|
| $e_1$ | A |
| $e_2$ | A |
| [...] | [...] |
| $e_{n-1}$ | B |
| $e_n$ | B |

Knowing that our model is already trained and composed of 1000 trees, each one being independent, each entry is given one after another to our model to predict its class. For each entry, each one of the 1000 trees will predict a class, the final decision given by the majority voting of all the 1000 trees. Knowing that each tree is independent, some may classify the entry of class A as class A (correct prediction), while others, based on their criteria, may misclassify the entry A as class B (wrong prediction). For instance: Given $T(A)$ the number of trees in the forest predicting an entry to be of class $A$ and $T(B)$ the number of trees in the forest predicting an entry to be of class $B$. Of note, in the actual computation, the number of trees is represented as floating number between 0 and 1. So 785 trees out of 1000 trees would actually equals 0.785 in the following formulas.

Thus, we establish the semantic: $T_B(A)_n$: *The number of trees that predicted class A for the entry n, knowing that the class of entry n is B*. And $N_B(A)$: *The number of entries predicted as class A knowing their class is B.*

Then, the CUP is computed as: with $CUP_B(A)$ the confidence of the model on the predictions that resulted in class A, knowing that the actual class is B:

$$CUP_B(A) = \frac{T_B(A)_1 + \ldots + T_B(A)_n}{N_B(A)}, N_B(A) > 0$$

So,

$$CUP_B(A) = \frac{\sum_{k=0}^{n} T_B(A)_k}{N_B(A)}, N_B(A) > 0$$

And,

$$CUP_B(A) = 0, N_B(A) = 0$$

In the case no entries of class B have been classified as A ($N_B(A) = 0$) then $CUP_B(A)$ equals to 0.

Given the fact that the CUP is a result of the majority voting by the trees of the Random Forest, we have a theoretical minimal value for the CUP computed as follow:

$$CUP_{min} = \frac{1}{N}$$

With $N$ the number of classes the model has been trained on, and $CUP_{min}$ the minimal CUP possible on a majority voting. If a prediction scores near the minimal CUP, it implies that the final decision was mostly random. Conversely, the closer to 1 is the CUP, the more trees voted for the final decision (given a single entry), and thus, the more reliable is the prediction. CUP equals to 0 occurs when no tree voted for a given class. In case of binary classification, the minimum CUP is 0.5 (50% of the trees voted for a certain class: random classification). A CUP of 0.9 means that an average of 90% of the trees on all entries voted for the final prediction. In summary, the more trees voted for a class, the most certain the model is about its decision.

We can also establish that, in case of binary classification (only class A and B):

$$T_B(B)_n + T_B(A)_n = 1$$

Considering a more general approach, given an entry $n$, and with $N$ the number of possible classes and $C_i$ one of those classes:

$$T_B(C_1)_n + T_B(C_2)_n + \ldots + T_B(C_N)_n = 1$$

$$\iff \sum_{k=1}^{N} T_B(C_k)_n = 1$$

In other terms, "for one entry of one class, the number of trees voting for all the possible classes sums up to 1".

### Accuracy of the model

The accuracy of the model is computed as:

$$\frac{\text{number of correct predictions}}{\text{total number of predictions}}$$

or, if given a binary classification:

$$\frac{TP + TN}{TP + TN + FP + FN}$$

Where TP: True Positives, TN: True Negatives, FP: False Positives, FN: False Negatives.

The maximum accuracy is 1, meaning that the model did no mistakes when predicting on a set of inputs. The theoretical minimum of the accuracy is computed as:

$$acc_{min} = \frac{1}{n_c}$$

With $n_c$ the number of classes the model has been trained on. If a model has an accuracy equivalent to the minimal theoretical accuracy, then it means that the model is giving random predictions.

### Cells and virus production

Vero E6 cells (CRL-1586) were obtained from the American Type Culture Collection and grown in DMEM (Sigma-Aldrich) supplemented with 10% fetal bovine serum (FBS, non-USA, Sigma-Aldrich) and 1% penicillin/streptomycin (P/S, Sigma-Aldrich). Tahyna virus (TAHV, NR-541) was obtained from the American Type Culture Collection (Virginia, WV, USA) through BEI Resources. Propagation and production of TAHV stocks was performed in Vero E6 cells. The cells were seeded in T175 flasks and incubated at 37 °C, 5% CO$_2$ to reach 80–90% confluency the following day, where the cells were absorbed with TAHV at a multiplicity of infection (MOI) of 0.0001 per cell (viral stock diluted in DMEM culture with 2% FBS, 1% P/S). The cells were incubated at 37 °C for 1 h, with gentle moving from side to side every 15 min. The inoculum was removed and the cells were incubated for 48–72 h at 37 °C with fresh DMEM, 2% FBS, 1% P/S. Virus was harvested by spinning the medium at 2000 × g for 10 min and harvesting the supernatant. Viral titers were determined by standard plaque assay in Vero E6 cells (described below) and measured as plaque forming units (PFU) per ml.

### LUHMES cell culture and differentiation

LUHMES cells were obtained from the Leist's lab and cultured and differentiated in monolayers as previously described (Lotharius et al, 2005; Scholz et al, 2011) with a few modifications. Briefly, for proliferation of undifferentiated monolayers, LUHMES cells were maintained at 37 °C, 5% CO$_2$ and 95% humidified atmosphere, in culture dishes previously coated with 50 µg/ml poly-L-ornithine and 1 µg/ml fibronectin. When differentiated on glass coverslips, 10 µg/ml laminin was added to the coating solution. Proliferation medium comprised Advanced DMEM/F12 medium supplemented with N2 supplement, 2 mM GlutaMAX, 40 ng/ml bFGF, and 1X penicillin-streptomycin. Cells were passaged every 2–3 days: they were detached using TrypLE, diluted in Advanced DMEM/F12 medium, centrifuged at 300 × g for 5 min and resuspended in proliferation medium. Cells were regularly tested for mycoplasma contamination.

For differentiation, LUHMES cells were first seeded in proliferation medium: 2.6 × 10$^6$ cells per 10 cm dish. The next day, cells were switched to differentiation medium (Advanced DMEM/F12 medium supplemented with N2 supplement, 2 mM GlutaMAX, 1 mM dbcAMP, 1 µg/ml tetracycline, 2 ng/ml GDNF, and 1X penicillin-streptomycin). After 48 h in differentiation medium, cells were detached, counted and seeded at a density of 1.6 × 10$^5$ cells/cm$^2$ in differentiation medium. After 5 days of differentiation, cells were washed with Advanced DMEM/F12 medium and incubated in tetracycline- and antibiotic-free differentiation medium for an additional 1–2 days before use as post-mitotic neurons.

### Drug treatment

The effect of the compounds on the viral infection in Vero E6 cells, LUHMES-derived neurons and OPAB was determined by measuring viral RNA levels by RT-qPCR. Vero E6 cells were seeded in 48-well flat-bottom plates at 30,000 cells/well and LUHMES-derived neurons were seeded in 24-well flat-bottom plates at 400,000 cells/well, and incubated at 37 °C for 24 h. Cells were pre-treated with BafA1, RG10b and Rottlerin at indicated concentrations. Vero E6 cells were infected with TAHV at a MOI of 0.1 for 1 h at 37 °C in the presence or absence of the compounds. The viral inoculum was then removed, the cell monolayer washed 2 times with PBS, fresh DMEM containing the compounds was added and plates incubated at 37 °C for indicated times.

For IC$_{50}$/CC$_{50}$ measurements, the cells were treated with a serial dilution of RG10b from 160 µM to 0.078 µM. The IC$_{50}$ was calculated from normalizing highest Ct value (RT-qPCR) and titer (Plaque assay) to equal 100% infection. The data were plotted as dose–effect curves fitted to a nonlinear regression model using the GraphPad Prism 9.5.1 software.

OPABs were placed in cellQART 12-well transwell inserts 0.4 µm PET (Sabeu, Northeim, Germany) with 1.2 ml N2 medium at the bottom of the well and infected with 10$^6$ PFU TAHV by placing it directly on top of the OPAB. The N2 medium and virus mix contained RG10b and Rottlerin at a final concentration of 10 µM and 20 µM, respectively. The OPABs were incubated with the virus at 37 °C for 4 h, the virus washed 2 times with N2 medium, any remaining liquid removed from the top of the transwell and plates placed back at 37 °C. Every 2–3 days the N2 medium at the bottom of the well was collected for cytotoxicity assay and fresh medium supplemented with the drugs was added. At each timepoint, the OPABs were washed 2 times with PBS and processed for RNA extraction using the RNeasy kit. Following the RLT addition, the OPABs were lysed by pipetting up and down, spun at max speed for 5 min, supernatant collected and proceeded to standard RNeasy protocol.

**The paper explained**

**Problem**

Viral infection of the brain is a major health concern worldwide. However, modeling the human brain *in vitro* is a very complex task, making the search for antivirals particularly challenging.

**Results**

We developed the culture of *post-mortem* human brain explants originating from autopsies and managed to maintain neural cells alive for weeks. Upon characterization of this model, we were able to infect them with a neurotropic virus in order to assess its impact on brain functions. We found that an antiviral molecule under development could inhibit viral replication, but was unable to fully prevent viro-induced perturbations of the electrical activity of the brain explant.

**Impact**

Here, we provide a proof-of-principle that *post-mortem* human brain explants can be a valuable resource for preclinical evaluation of antiviral molecules, with the unique advantage to account for human-to-human variability of responses.

## RT-qPCR

At indicated timepoints, cells were washed 2 times with PBS and processed for RNA extraction using the RNeasy kit (Qiagen, Hilden, Germany). TAHV RNA levels were measured utilizing the Luna® Universal One-Step RT-qPCR kit, using the following conditions: 25 °C for 2 min; reverse transcription step at 50 °C for 15 min; initial denaturation step at 95 °C for 2 min; 40 cycles of denaturation at 95 °C for 15 s and extension at 55 °C for 30 s. TAHV-specific primers targeted the M segment and the RNA levels were normalized to Glyceraldehyde 3-phosphate dehydrogenase (GAPDH) using the following primers: TAHVFor GGTCCTA CATTGCCGTTCAAG, TAHVRev TGGTCTACAGGTGCTAGC TC, GAPDHFor GAGTCAACGGATTTGGTCGT and GAPDHRev TTGATTTTGGAGGGATCTCG. The RT-qPCR reaction was performed on a Mx3000P qPCR system (Agilent, Santa Clara, California, USA).

## Plaque assay

Vero E6 cells were seeded in 12-well plates at 450,000 cells/well and incubated at 37 °C for 24 h. A series of 10-fold dilutions of virus was prepared in DMEM (2% FBS, 1% P/S) in 96-well round bottom plate, starting with 225 µl medium + 25 µl virus ($10^{-1}$ dilution). The medium from a confluent monolayer of VeroE6 cells was removed and cells inoculated with 200 µl of desired dilutions per well. The virus was adsorbed for 1 h at 37 °C with gentle plate rocking every 15 min. A 1 ml overlay (for 20 ml; 2 ml DMEM 10X, 1 ml FBS, 320 µl 7.5% NaHCO₃, 100 µl 1000X P/S, 500 µl, 1 M Hepes, 200 µl 100X Glutamine, 10.88 ml sterile H₂O and 5 ml melted 1% agarose in H₂O) was placed per well and plates incubated at 4 °C for 10 min to allow agarose to solidify. The plates were subsequently incubated at 37 °C for 3 days and cells fixed by adding 1 ml of 4% PFA and incubated for 30 min at room temperature. The PFA and overlays were removed from the wells, cells stained with 500 µl Crystal Violet (0.1% v/v in 10% EtOH) per well and incubated for 1 h with gentle rocking. The Crystal Violet was removed, rinsed clear with water, plates dried and plaques counted. The viral titer was determined using the following formula: No. of Plaques/(Dilution Factor × Volume of diluted virus in ml/well) = PFU/ml.

## Statistics

Data were statistically analyzed using student t tests, or confidence upon prediction (CUP) as indicated in the figure legends. T tests were calculated using Graphpad Prism 9.5.1 (or above) or Excel 2019 software.

## Data availability

The codes used for machine learning analyses are available on Github: https://github.com/WillyLutz/tahynavirus-electrical-analysis.

## Peer review information

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

## Acknowledgements

LUHMES cells were a generous gift from Stefan Schildknecht from Marcel Leist's laboratory in Konstanz, Germany. Image acquisition was performed at the MRI imaging facility (CNRS, Univ Montpellier), which also provided advice and training. We thank Valérie Rigau for helpful discussions, Corentin Bernou for participating in OPAB preparation, and Marie-José Partiot for the scientific illustrations. We thank the CRB of the CHU Montpellier for processing the IHC. The following reagent was obtained through BEI Resources, NIAID, NIH: Tahyna Virus, 92, NR-541. Agence Nationale de la Recherche ANR CIBORG (ANR-21-CE33-0007-03) to GG, SC, and RG; ANR MigraVir (ANR-20-CE15-0019-01) to RG and ANR isiBrain (ANR-22-CE15-0007-01) to RG and SC. Fondation pour la Recherche Médicale (FRM) – ANRS-MIE (MIE202207016212) to EP and RG. Isite MUSE (Montpellier University of Excellence) to BC.

## Author contributions

**Emma Partiot**: Formal analysis; Validation; Investigation; Visualization; Methodology. **Barbara Gorda**: Formal analysis; Validation; Investigation; Visualization; Methodology. **Willy Lutz**: Resources; Software; Formal analysis; Validation; Investigation; Visualization; Methodology. **Solène Lebrun**: Formal analysis; Investigation; Methodology. **Pierre Khalfi**: Investigation; Methodology; Writing—review and editing. **Stéphan Mora**: Investigation; Methodology. **Benoit Charlot**: Methodology. **Karim Majzoub**: Supervision; Methodology; Writing—review and editing. **Solange Desagher**: Supervision; Investigation; Methodology; Writing—review and editing. **Gowrishankar Ganesh**: Conceptualization; Software; Formal analysis; Supervision; Funding acquisition; Validation; Investigation; Visualization; Methodology; Writing—review and editing. **Sophie Colomb**: Conceptualization; Resources; Formal analysis; Supervision; Funding acquisition; Investigation; Methodology; Writing—review and editing. **Raphael Gaudin**: Conceptualization; Resources; Formal analysis; Supervision; Funding acquisition; Validation; Investigation; Visualization; Methodology; Writing—original draft; Writing—review and editing.

## Disclosure and competing interests statement

The authors declare no competing interests.

# Expanded View Figures

**Figure EV1.  Immunohistochemistry of OPAB.**

(**A**, **B**) Immunochemistry of Hemalin-Eosin (H&E), neurons (NeuN), oligodendrocytes (Oligo2) and astrocytes (GFAP) from parietal (**A**) or frontal (**B**) areas of OPAB at indicated days in vitro (DIV). The condition 0 DIV correspond to immediate sample collection at autopsy, before vibratome slicing and OPAB. The black squares correspond to the zoomed area on the right panels. Images were acquired on Leica Thunder using 20x magnification. Scale bar: 100 μm.

▶

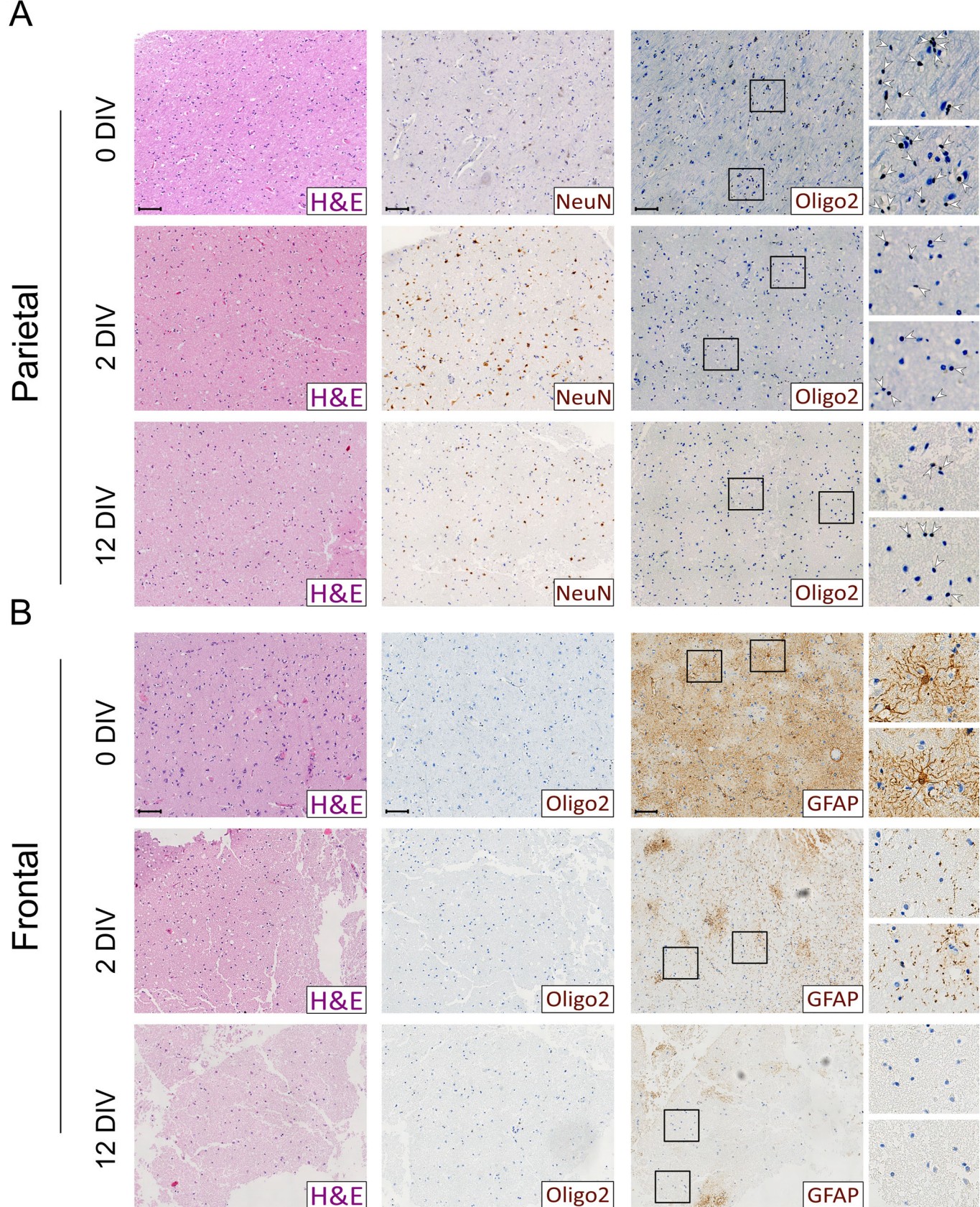

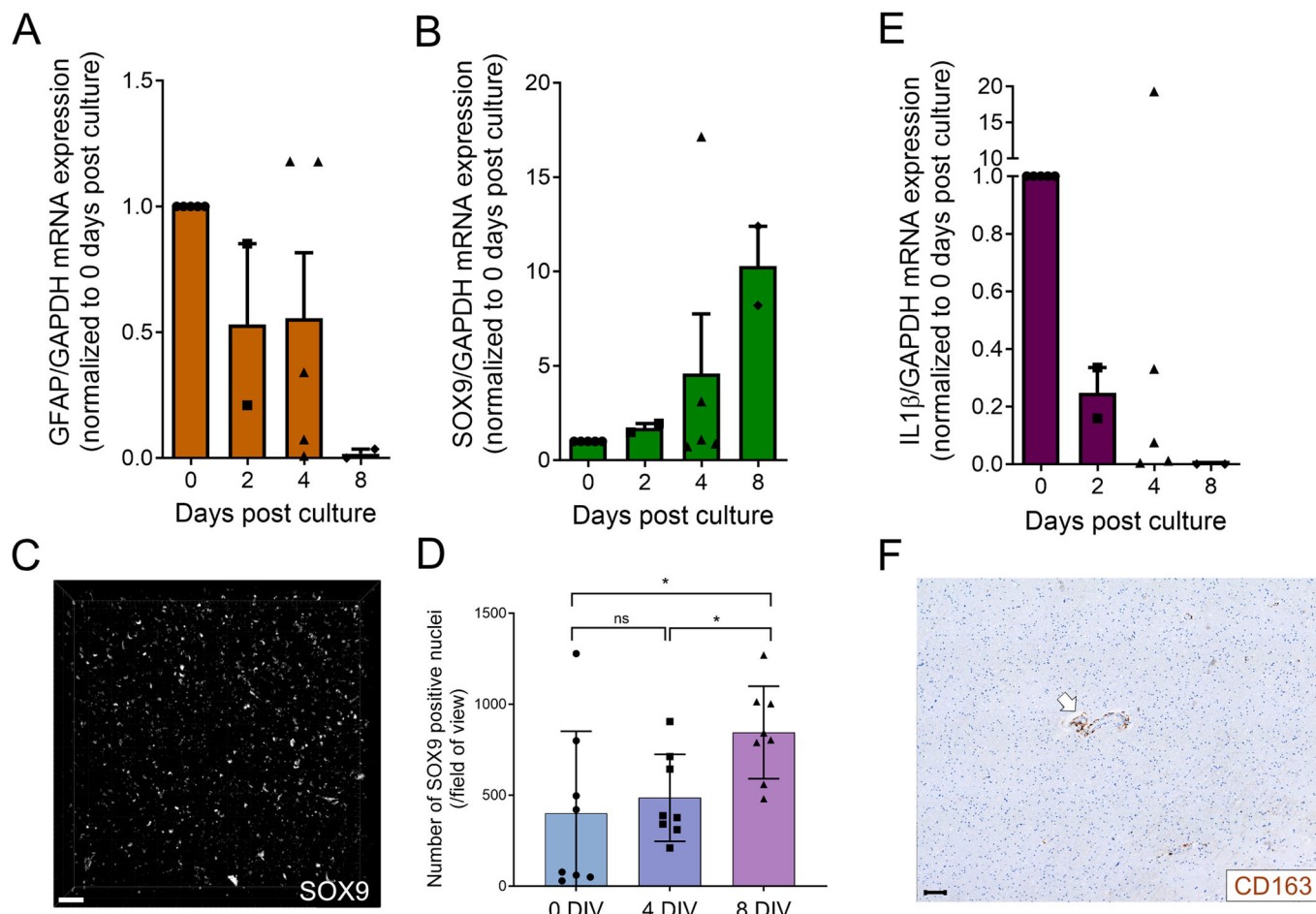

**Figure EV2. Characterization of astrocytic and immunoreactivity of OPAB.**

(A, B) OPAB were cultured for indicated days in vitro, RNA was extracted and RT-qPCR was performed to measure the expression levels of GFAP- (A) and SOX9-coding (B) mRNA over time. Data are mean $+/-$ SEM and each dot corresponds to the measurement of the mRNA levels per slice, coming from at least two donors. (C, D) OPAB were fixed at 4 DIV (C) or indicated times. (D) and stained with anti-SOX9 antibody, labeling the nucleus of astrocytes. (C) Representative 3D confocal image of SOX9 staining. Scale bar: 50 μm. (D) Quantification of the number of SOX9-positive nuclei per field of view at indicated time post in vitro culture. Unpaired t-test p value < 0.05 (*); ns: non-significant. Data are mean $+/-$ SD from 8 fields of view taken from two slices from two donors. (E) Samples processed as in A-B were assessed for expression of IL1β-coding mRNA. Of note, one slice at day 4 post culture exhibit oddly high values preventing reliable statistical analysis. Data are mean $+/-$ SEM and each dot corresponds to the measurement of the mRNA levels per slice, coming from at least two donors. (F) Immunohistochemistry of OPAB at 0 DIV showing absence of monocyte/macrophage/microglia (CD163, brown) staining in the cortex. The white arrow highlights a blood vessel at which CD163 staining is elevated, indicating that monocyte/macrophages did not extensively infiltrated the cortex.

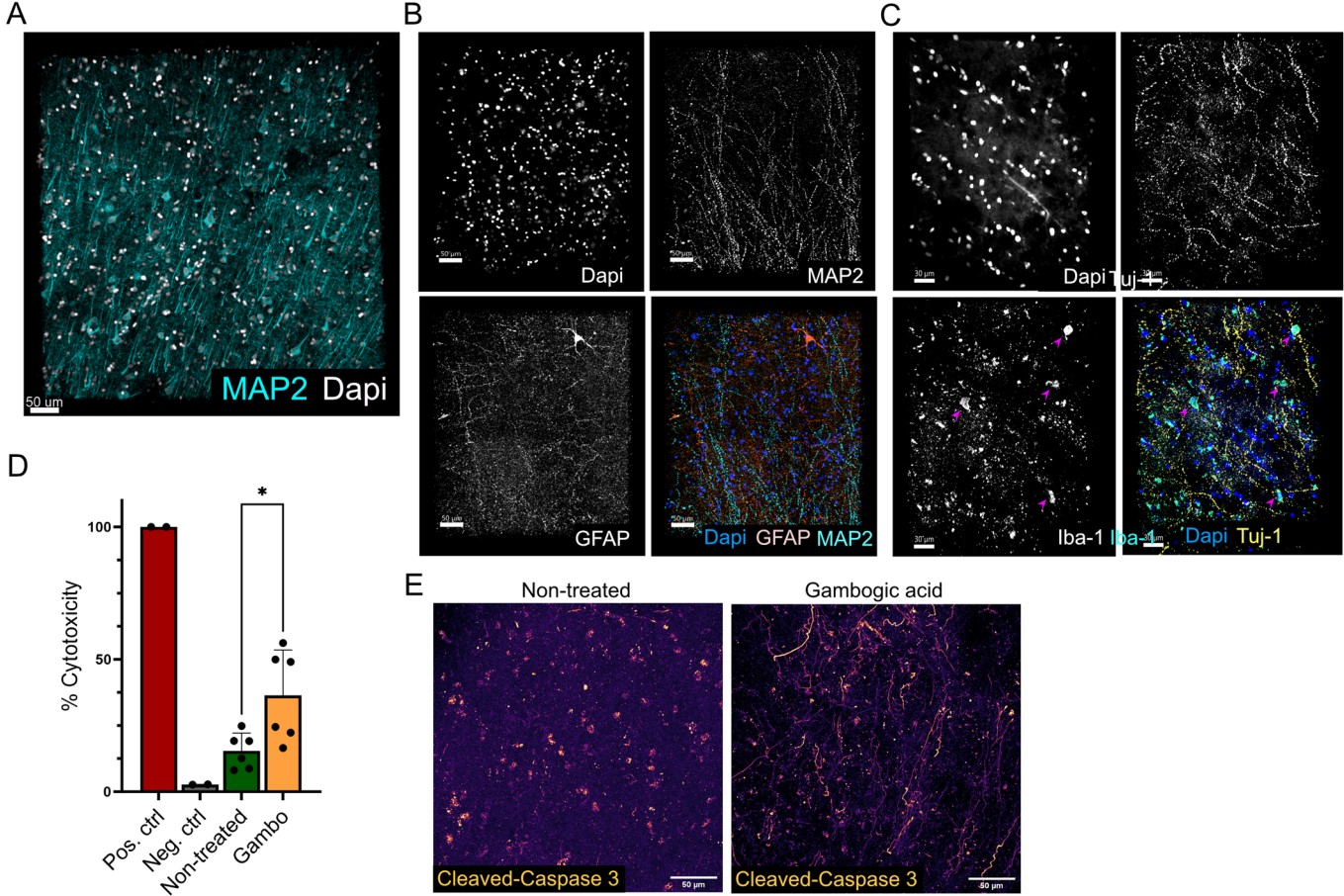

Figure EV3. Characterization of OPAB by immunofluorescence.

(A) Snapshots from 3D confocal imaging (top view) of frontal OPAB, showing parallelly aligned post-mitotic neurons (MAP2 in cyan) and nuclei (Dapi in white). Scale bar: 50 μm. (B, C) Three-dimensional imaging of neurons (Tuj1 and MAP2), astrocytes (GFAP) and microglial cells (Iba1) from a frontal OPAB at 5 DIV. (D) Neuronal LUHMES cells treated for 24 h with 10 μg/ml Gambogic acid (Gambo), and cytotoxicity was measured in the supernatant using LDH assay. Unpaired t-test $p$ value < 0.05 (*). Data are mean $+/-$ SD from 2 individual experiments performed in triplicates. Controls were performed once per experiment. (E) OPAB were treated for 24 h with 10 μg/ml Gambogic acid and anti-Cleaved-Caspase-3 antibody labeling was performed. Confocal snapshots highlight brighter Cleaved-Caspase-3 staining upon Gambogic acid treatment. Scale bar: 50 μm.

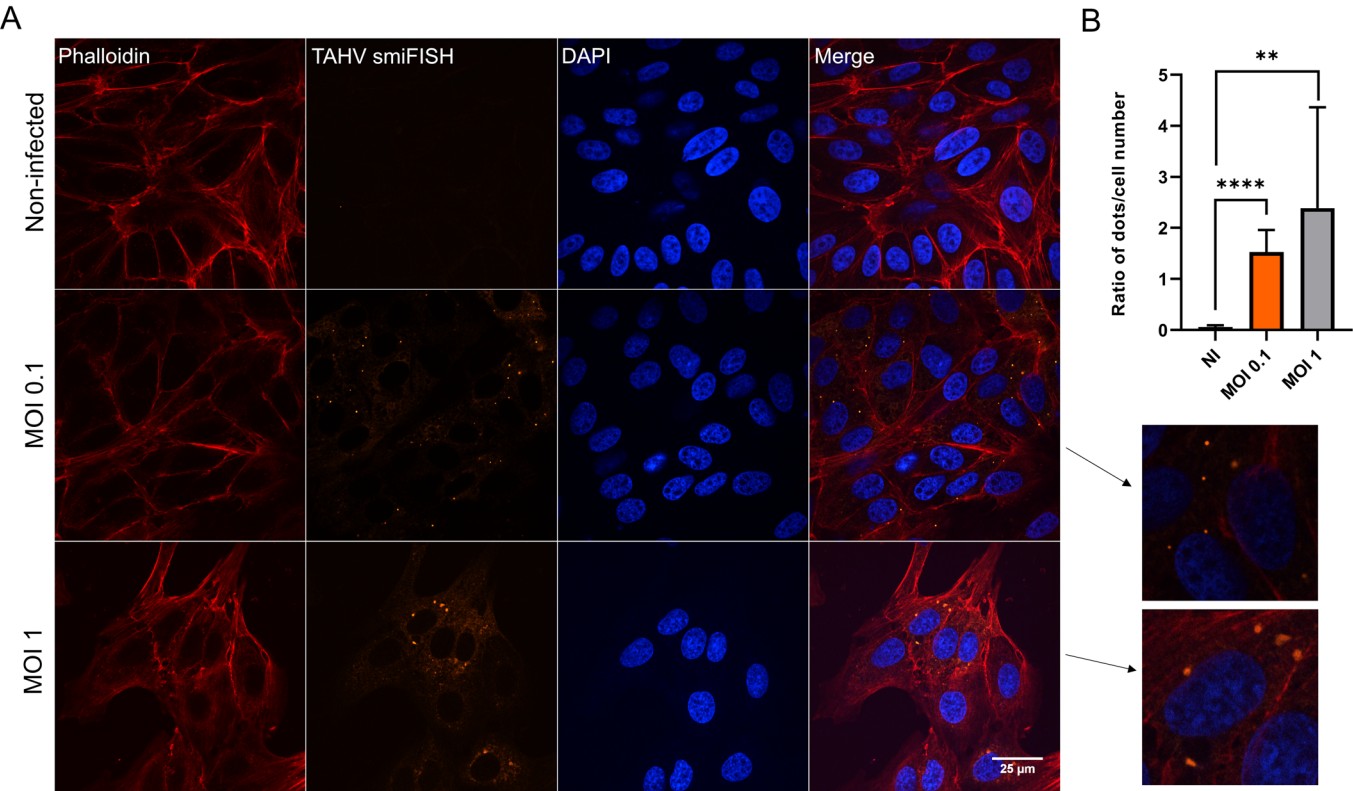

**Figure EV4. Characterization of TAHV smFISH in Vero E6 cells.**

(A) Vero E6 cells were infected with TAHV for 24 h at indicated MOI and stained with phalloidin (actin, red), TAHV smFISH (viral RNA, orange), and Dapi (nuclei, blue). Scale bar: 25 μm. Lower right panels correspond to two magnified crops from indicated conditions, highlighting dotted structures likely corresponding to viral factories. (B) Quantification of the number of large dotted structures from the TAHV smFISH staining in (A). Data are mean $+/-$ SD from 10 fields of view per condition from an experiment. Unpaired t-test $p$ value < 0.01 (**) or <0.0001 (****).

