## [Peer Review File · EMBO Molecular Medicine]

Organotypic culture of human brain explants as a preclinical model for AI-driven antiviral studies

Emma Partiot, Barbara Gorda, Willy Lutz, Solène Lebrun, Pierre Khalfi, Stéphan Mora, Benoit Charlot, Karim Majzoub, Solange Desagher, Gowrishankar Ganesh, Sophie Colomb, and Raphael Gaudin

Corresponding author: Raphael Gaudin (raphael.gaudin@irim.cnrs.fr)

Review Timeline:

Submission Date:	5th Sep 23
Editorial Decision:	11th Oct 23
Revision Received:	12th Jan 24
Editorial Decision:	25th Jan 24
Revision Received:	2nd Feb 24
Accepted:	5th Feb 24

Editor: Zeljko Durdevic

Transaction Report:

11th Oct 2023

Dear Dr. Gaudin,

Thank you for the submission of your manuscript to EMBO Molecular Medicine. We have now received feedback from the three reviewers who agreed to evaluate your manuscript. All three referees recognize potential interest of the study but also raise serious concerns that should be addressed in a major revision. Adding more donors as suggested by the referee #3 is welcomed but not required for further consideration. If you would like to discuss further the points raised by the referees, I am available to do so via email or video. Let me know if you are interested in this option.

We would welcome the submission of a revised version within three months for further consideration. Please let us know if you require longer to complete the revision.

Please use this link to login to the manuscript system and submit your revision: <https://embomolmed.msubmit.net/cgi-bin/main.plex>

I look forward to receiving your revised manuscript.

Yours sincerely,

Zeljko Durdevic

We require:

- 1) A .docx formatted version of the manuscript text (including legends for main figures, EV figures and tables). Please make sure that the changes are highlighted to be clearly visible.
- 2) Individual production quality figure files as .eps, .tif, .jpg (one file per figure). For guidance, download the 'Figure Guide PDF': (<https://www.embopress.org/page/journal/17574684/authorguide#figureformat>).
- 3) A .docx formatted letter INCLUDING the reviewers' reports and your detailed point-by-point responses to their comments. As part of the EMBO Press transparent editorial process, the point-by-point response is part of the Review Process File (RPF), which will be published alongside your paper.
- 4) A complete author checklist, which you can download from our author guidelines (<https://www.embopress.org/page/journal/17574684/authorguide#submissionofrevisions>). Please insert information in the checklist that is also reflected in the manuscript. The completed author checklist will also be part of the RPF.
- 5) Please note that all corresponding authors are required to supply an ORCID ID for their name upon submission of a revised manuscript.
- 6) It is mandatory to include a 'Data Availability' section after the Materials and Methods. Before submitting your revision, primary

datasets produced in this study need to be deposited in an appropriate public database, and the accession numbers and database listed under 'Data Availability'. Please remember to provide a reviewer password if the datasets are not yet public (see <https://www.embopress.org/page/journal/17574684/authorguide#dataavailability>).

13) Author contributions: You will be asked to provide CRediT (Contributor Role Taxonomy) terms in the submission system. These replace a narrative author contribution section in the manuscript.

14) A Conflict of Interest statement should be provided in the main text.

Please note: When submitting your revision you will be prompted to enter your funding and payment information. This will allow Wiley to send you a quote for the article processing charge (APC) in case of acceptance. This quote takes into account any reduction or fee waivers that you may be eligible for. Authors do not need to pay any fees before their manuscript is accepted and transferred to the publisher.

EMBO Press participates in many Publish and Read agreements that allow authors to publish Open Access with reduced/no publication charges. Check your eligibility: <https://authorservices.wiley.com/author-resources/Journal-Authors/open-access/affiliation-policies-payments/index.html>

***** Reviewer's comments *****

Referee #1 (Comments on Novelty/Model System for Author):

The combination of AI and slice culture of human brain is a novel method for analyzing alteration of network activity patterns in adult human neural network. It can be utilized in diverse field if it becomes more accessible for more researchers.

Referee #1 (Remarks for Author):

The manuscript describes alteration of network activity patterns of human adult brain slice culture upon neurotropic viral infection. It demonstrates that the infection can be precisely monitored by "AI" algorithms, although the activity patterns of brain slices from multiple donors are different.

The "AI" algorithm used in the study was random forest. It was unclear from the manuscript that RFC was the best choice among many AI algorithms. More explanation and/or exploration would be expected, given the title of the manuscript contains "AI".

LFP spectrum (e.g. Figure 2F, 4C) show high frequency signal components which reflect mostly spikes (unless noise). If LFP is 2000Hz, it would be oscillation components with a period of 0.5 millisecond. Would this be meaningful for comprehending neuronal activity patterns? Low frequency range (e.g. under 300Hz) should be also considered for analysis.

Referee #2 (Comments on Novelty/Model System for Author):

The generalization of such as model system would require completing several practical and regulatory steps, which may not be easy to achieve in all laboratories. Indeed, it would require proximity and close interactions with a forensics department, to ensure that brain harvesting could be performed with a postmortem interval inferior to 12 hours (as shown here), followed by a quick transfer to the lab. Accessing to such a material also would require formal approval by regulatory agencies, which may not be easy to obtain in all countries. Finally, there are in my view several not fully resolved issues in the manuscript under its current form, notably about the actual viability of the samples and of their permissivity to infections. This is why I believe that the adequacy of the model system is unclear at this stage.

Referee #2 (Remarks for Author):

Review

In this work, the authors describe a model of organotypic cultures of human brain explants, prepared using post-mortem brain samples (collected with 12 hours after death and rapidly processed thereafter). Interestingly, they provide evidence that these cultures exhibit an overall conservation of the cortical organization and synaptic connectivity for at least two weeks. They also show that they can detect electrical activity of their samples (in the form of Local Field Potentials, or LFP), using an original MEA system.

The authors then move on to demonstrate that their samples can be infected by a model virus, leading to changes in the LFP patterns and that some antiviral molecules can interfere with the virus and restore a "normal" LFP pattern. For the analysis of the

LFP patterns, the authors have designed an AI-driven tool for their automated analysis.

Although several questions remain, I found that this work was very carefully done and overall convincing. There are, however, several aspects that the authors may want to consider.

General points:

1. In the presentation of their manuscript, there is a slight tendency of "overselling" the rationale and potential impact of the study. In my view, this is unnecessary considering the quality of the supporting data and it may even be counter-productive. For instance, there are mentions in the abstract (Lines 32-33) and also in introduction about the pro and cons of different neural models and on their suitability to study cognitive functions. As a matter of fact, the model proposed by the authors does not provide any added value in this matter and thus there is no need in my view to discuss this matter.

2. In their assays to study neuronal connectivity, the authors show convincing pictures of synaptic structures, although it is difficult to appreciate the actual extent of this synaptic connectivity in the whole sample. In that respect, measuring electrical activity is indeed a very logical and appropriate method. However, the authors conclude that "this model is suitable for neural network studies", based on the sole detection of LFP, whose origin, relevance and role are not clearly known. To fully validate the electrical viability of their samples, the authors could consider showing (1) loss of electrical signals upon exposure to reversible silencing agents (such as Tetrodotoxin, in combination with NMDA inhibitors); (2) likewise, they could show enhanced electrical activity upon depolarization (using high KCl). These experiments would definitely prove that the spontaneous LFP that they measure are a bona fide surrogate marker of neuronal activity of living neurons at steady-state.

3. Lines 126 to 129: I am not entirely convinced by the interpretation for the very rapid drop observed in GFAP staining. Reactive astrogliosis consecutive the axotomy (during cutting) is indeed expected, but usually such a gliotic "scar" takes a long time to disappear, contrasting with the rapid drop that the authors observe. Such a rapid drop could alternatively result for death and loss of the glial cells. Do the authors have any evidence (TUNEL for instance) that would show that there is no cell loss in their samples? I am also not entirely convinced by the interpretation for the absence of T lymphocytes. Sample collection was performed post-mortem, therefore without any blood flow to irrigate the tissue. Thus, I don't think that the authors can talk of "immune-activation" consecutive to collection. In my view, the absence of overt T lymphocyte in the brain parenchyma in an indirect proof that the patient was not suffering from massive neuroinflammation before his/her death.

4. Line 147: The authors used Gambogic acid, which is presumed to be a neurotoxic compound. However, details are lacking about the neurotoxicity of this compound in the cited reference, which rather focuses on other neurotoxins. Therefore, the rationale for choosing this particular drug, whose effect is not well understood, is not explained with sufficient clarity. Have the authors considered experiments using Gambogic acid treatments of cultured or primary neuronal cultures to formally demonstrate its neurotoxic activity in a more "controlled" setting?

5. The authors need to better justify their use of Tahyna virus (THAV, which incidentally should be spelt in full the first time it cited in the manuscript on line 84). To justify their choice, the authors state that several orthobunyaviruses belonging to the California subgroup (CSG), including THAV, were recently shown as being associated to CNS diseases. However, when examining closely the bibliography cited by the authors, it appears that evidence for THAV being neurotropic is scarce and inconclusive: findings are based only on serological data and detection of this virus has never been reported in the CNS. Despite RNA detection by RT-PCR, there is no other proof provided by the authors that the virus is actually infecting and replicating in the brain slices, in particular in neurons. It would be important to show the presence of viral antigen in the tissues. I understand that antibodies to THAV may be difficult to obtain, but is there any possibility to use antibodies cross-reacting with several members of this viral family, or to perform RNA-FISH experiments to demonstrate viral presence in the tissue? Another important missing control would be to perform control infections using UV-irradiated virus to formally exclude that the RNA detection results would only represent a slow decay of the viral inoculum, or "peripheral" low level virus replication. Another element that would give a good indication of the ability of THAV to infect neural cells would be to document the ability to achieve in vitro infections using a human cell line of neural/neuronal origin, such as SHSY5Y or U-373MG cells (non-limitative list).

6. Although it discussed later in the manuscript, the rationale for testing the antiviral potential of RG10b is not explained with sufficient clarity. The authors indicate that they previously demonstrated the efficacy of this molecule on Coronaviruses, but this may not be a very valid argument. Indeed, CoVs are positive stranded RNA viruses, whereas THAV is a segmented negative-stranded RNA. Rather, the authors could indicate that the putative mechanisms of action of RG10b (which is still not fully understood, but may be linked to perturbations at the level of the ER) may be applicable to bunyaviruses.

Other points:

1. The authors need to be consistent and more careful about their use of tenses. In particular, the results and methods section should be written in the past tense throughout.

2. The authors should indicate the agreement numbers issued by the agencies that approved the study.

3. "media" should be replaced by "medium" (see for instance lines 366 to 376, or 430 to 437).

4. I am not sure that it is really necessary to abbreviate "air-liquid interface" by ALI

Referee #3 (Comments on Novelty/Model System for Author):

The concept taken by authors are novel in the field of virology and its related domains.

Referee #3 (Remarks for Author):

The manuscript is well conceptualize and novel in the field of virology. However there are three donors were included in this study. Data variation among these three donors are prominent which makes the hypothesis weak. Therefore, adding more donors in experimental setup may enhance the power of study and relevance of hypothesis strongly.

Point-by-point response to reviewers' comments (EMM-2023-18646)

Referee #1 (Comments on Novelty/Model System for Author):

The combination of AI and slice culture of human brain is a novel method for analyzing alteration of network activity patterns in adult human neural network. It can be utilized in diverse field if it becomes more accessible for more researchers.

We thank the reviewer for point out the novelty of our work.

Referee #1 (Remarks for Author):

The manuscript describes alteration of network activity patterns of human adult brain slice culture upon neurotropic viral infection. It demonstrates that the infection can be precisely monitored by "AI" algorithms, although the activity patterns of brain slices from multiple donors are different. The "AI" algorithm used in the study was random forest. It was unclear from the manuscript that RFC was the best choice among many AI algorithms. More explanation and/or exploration would be expected, given the title of the manuscript contains "AI".

We thank the reviewer for his/her rightful comment. We tested RFC along with SVM and SGDR during the early stages of the algorithm's development, and found that RFC was providing far more compelling results than SVM and SGDR. RFC has also the advantage to return feature weights to pinpoint the contribution of each frequency in prediction outcome. Although other types of machine learning strategies may also perform well on LFP analyses, this work highlights the promise of how electrical activity may be unbiasedly analyzed using AI. Accordingly, we have added these details to explain the choice of RFC in the Discussion.

LFP spectrum (e.g. Figure 2F, 4C) show high frequency signal components which reflect mostly spikes (unless noise). If LFP is 2000Hz, it would be oscillation components with a period of 0.5 millisecond. Would this be meaningful for comprehending neuronal activity patterns? Low frequency range (e.g. under 300Hz) should be also considered for analysis.

As suggested by the reviewer, we ran analyses in which we considered only the lower frequencies (0-1000Hz). As shown in the figure below, the trends remain the same for all three donors, but accuracy was lower due to less datapoints taken into consideration. Hence, we chose to keep the whole frequency spectrum, as the AI algorithm still finds some important features associated to high frequencies. We agree that the meaning of high frequency data is unclear, but the ways AI is working goes sometimes beyond rational explanations...

Referee #2 (Comments on Novelty/Model System for Author):

The generalization of such as model system would require completing several practical and regulatory steps, which may not be easy to achieve in all laboratories. Indeed, it would require proximity and close interactions with a forensics department, to ensure that brain harvesting could be performed with a postmortem interval inferior to 12 hours (as shown here), followed by a quick transfer to the lab. Accessing to such a material also would require formal approval by regulatory agencies, which may not be easy to obtain in all countries. Finally, there are in my view several not fully resolved issues in the manuscript under its current form, notably about the actual viability of the samples and of their permissivity to infections. This is why I believe that the adequacy of the model system is unclear at this stage.

Referee #2 (Remarks for Author):

In this work, the authors describe a model of organotypic cultures of human brain explants, prepared using post-mortem brain samples (collected with 12 hours after death and rapidly processed thereafter). Interestingly, they provide evidence that these cultures exhibit an overall conservation of the cortical organization and synaptic connectivity for at least two weeks. They also show that they can detect electrical activity of their samples (in the form of Local Field Potentials, or LFP), using an original MEA system.

The authors then move on to demonstrate that their samples can be infected by a model virus, leading to changes in the LFP patterns and that some antiviral molecules can interfere with the virus and restore a "normal" LFP pattern. For the analysis of the LFP patterns, the authors have designed an AI-driven tool for their automated analysis.

Although several questions remain, I found that this work was very carefully done and overall convincing. There are, however, several aspects that the authors may want to consider.

We thank the reviewer for acknowledging the thoroughness of our work.

General points:

1. In the presentation of their manuscript, there is a slight tendency of "overselling" the rationale and potential impact of the study. In my view, this is unnecessary considering the quality of the supporting data and it may even be counter-productive. For instance, there are mentions in the abstract (Lines 32-33) and also in introduction about the pro and cons of different neural models and on their suitability to study cognitive functions. As a matter of fact, the model proposed by the authors does not provide any added value in this matter and thus there is no need in my view to discuss this matter.

We agree with the reviewer that we have not proven that this model is suitable to study cognitive functions at this stage. Thus, we have deleted our references to neurocognition in the abstract and introduction. We also added a sentence in the Discussion to question whether our model could indeed be used to study cognitive functions, as a perspective of research.

2. In their assays to study neuronal connectivity, the authors show convincing pictures of synaptic structures, although it is difficult to appreciate the actual extent of this synaptic connectivity in the whole sample. In that respect, measuring electrical activity is indeed a very logical and appropriate

method. However, the authors conclude that "this model is suitable for neural network studies", based on the sole detection of LFP, whose origin, relevance and role are not clearly known. To fully validate the electrical viability of their samples, the authors could consider showing (1) loss of electrical signals upon exposure to reversible silencing agents (such as Tetrodotoxin, in combination with NMDA inhibitors); (2) likewise, they could show enhanced electrical activity upon depolarization (using high KCl). These experiments would definitely prove that the spontaneous LFP that they measure are a bona fide surrogate marker of neuronal activity of living neurons at steady-state.

We thank the reviewer for making this important remark. To tone-down our claim, we modified the conclusion by stating that this model could be suitable for neural network studies.

Furthermore, following reviewer's suggestion we treated OPAB with electrical inhibitors, showing a decrease of the number of spike detection. This treatment is highly variable, although it overall significantly decreases LFP (* p value < 0.05). This variability is not surprising because we are looking at spontaneous electrical activity, and no attempt to synchronize the signal was undertaken in this study. This could be achieved with electrical stimulation, but it would require massive amount of work and optimization, which we believe are falling beyond the scope of this work. Moreover, we also performed KCl experiments as requested by the reviewer, but we have been facing important technical challenges that could not be overcome in such short time. Indeed, addition of KCl induces immediate electrical responses, but electrical activity needs to be measured in a shielded environment, and actually, just the pipetting was already affecting the background of the electrical activity. A microfluidic system would be needed to allow us to measure electrical activity within the aluminum shield and without pipetting-induced perturbations within milliseconds post-KCl addition. We are currently developing such microfluidic engineering, but this is a complex and sensitive approach that we do not have controlled well-enough for now.

3.Lines 126 to 129: I am not entirely convinced by the interpretation for the very rapid drop observed in GFAP staining. Reactive astrogliosis consecutive the axotomy (during cutting) is indeed expected, but usually such a gliotic "scar" takes a long time to disappear, contrasting with the rapid drop that the authors observe. Such a rapid drop could alternatively result for death and loss of the glial cells. Do the authors have any evidence (TUNEL for instance) that would show that there is no cell loss in their samples?

We thank the reviewer for this remark, who raised an important question that was not addressed in our previous version of the manuscript. We have now added in the new **Expanded View EV21A-D** evidences that GFAP disappearance was not due astrocytic death, but rather to lower immunoreactivity. Indeed, we quantified and confirmed the GFAP expression decrease over time in OPAB (**Expanded View EV2A**). Moreover, we monitored SOX9 expression, a nuclear astrocytic marker that is independent of immune activation, and showed that the OPAB did not lose, but rather slightly increased SOX9 expression (**Expanded View EV2B-D**), indicating that astrocytes are still present after 8 days of culture, and could even maybe divide, which would explain the increased SOX9 levels (although this is too preliminary to be asserted at this stage).

I am also not entirely convinced by the interpretation for the absence of T lymphocytes. Sample collection was performed post-mortem, therefore without any blood flow to irrigate the tissue. Thus, I don't think that the authors can talk of "immune-activation" consecutive to collection. In my view, the absence of overt T lymphocyte in the brain parenchyma in an indirect proof that the patient was not suffering from massive neuroinflammation before his/her death.

We thank the reviewer for this comment. As proposed, we changed the interpretation of the absence of T cell infiltration by “suggesting that the explants did not experience neuroinflammation prior to the patient’s death.”. Moreover, we reinforced this statement by adding CD45 and CD163 stainings (new **Figure 1D** and **Expanded View EV2F**). Finally, we started to address the question of immune reactivity upon organotypic culture by monitoring the proinflammatory cytokine IL1b mRNA over time (new **Expanded View EV2E**). We saw a trend of decreased levels of IL1b, except for one slice. Together, we believe that these additional data reinforce the manuscript by providing a greater characterization of the system.

4.Line 147: The authors used Gambogic acid, which is presumed to be a neurotoxic compound. However, details are lacking about the neurotoxicity of this compound in the cited reference, which rather focuses on other neurotoxins. Therefore, the rationale for choosing this particular drug, whose effect is not well understood, is not explained with sufficient clarity. Have the authors considered experiments using Gambogic acid treatments of cultured or primary neuronal cultures to formally demonstrate its neurotoxic activity in a more "controlled" setting?

As suggested by the reviewer, we have added in new **Expanded View EV3D-E** proof of the cytotoxic activity of Gambogic acid on neurons and OPAB. We also described it in more details in the Results section.

5.The authors need to better justify their use of Tahyna virus (THAV, which incidentally should be spelt in full the first time it cited in the manuscript on line 84). To justify their choice, the authors state that several orthobunyaviruses belonging to the California subgroup (CSG), including THAV, were recently shown as being associated to CNS diseases. However, when examining closely the bibliography cited by the authors, it appears that evidence for THAV being neurotropic is scarce and inconclusive: findings are based only on serological data and detection of this virus has never been reported in the CNS.

The literature is indeed scarce on TAHV in humans, due to the low pathogenesis known to be associated of this specific virus. Hence, very few labs have searched for TAHV in the brain of deceased patients. However, closely-related viruses, such as LACV, are well known viruses that enter the CNS and cause important encephalitis in children. We agree that the clinical impact of TAHV in humans is unclear, but a study that we cited, showed that TAHV can reach the brain of immunocompetent mice. On a more personal note, we have tested OPAB infection with 4 neurotropic viruses, and TAHV seems to be the most infectious one. We cannot add these data because they have not been well-controlled so far and might fall beyond the scope of this study.

However, we might not have specified clearly enough the neurotropic potential of TAHV and thus we have now modified the introduction to detail a bit more the known facts about TAHV neuroinfection.

Despite RNA detection by RT-PCR, there is no other proof provided by the authors that the virus is actually infecting and replicating in the brain slices, in particular in neurons. It would be important to show the presence of viral antigen in the tissues. I understand that antibodies to THAV may be difficult to obtain, but is there any possibility to use antibodies cross-reacting with several members of this viral family, or to perform RNA-FISH experiments to demonstrate viral presence in the tissue?

We did test the pan-orthobunyavirus antibody from Santa Cruz (ref sc-58098), but it did not work for immunofluorescence nor Western blot, and has been discontinued since then.

Nevertheless, we developed smFISH assay to image the presence of the virus in neurons. Unfortunately, this technique was not successful in 3D tissues, probably because of the lack of accessibility of the probes in the OPAB, and the relatively high background observed in OPAB even in the absence of smFISH probes. Therefore, we used a neuronal cell line, the LUHMES cells, and showed that TAHV replicates efficiently in neurons using smFISH and RT-qPCR (**Figure 4E-F**). We also confirmed that RG10b is efficient in this model and that viral RNA in the inoculum is negligible by using UV-inactivated TAHV in neurons and OPAB (**Figure 4F** and **Figure 5B**).

Another important missing control would be to perform control infections using UV-irradiated virus to formally exclude that the RNA detection results would only represent a slow decay of the viral inoculum, or "peripheral" low level virus replication.

Although we understand the concern of the reviewer about presence of viral RNA from the inoculum, the fact that we do not see a decay, but an increase of the viral RNA over time clearly indicates that the virus replicates, and that we are not only detecting the inoculum. Nevertheless, as suggested by the reviewer, we have now included data with a UV-inactivated virus and showed that indeed, UV-irradiated TAHV is unable to replicate in brain slices and that background RNA levels are found, potentially coming from the viral input. These data have been added in new **Figure 5B** of the revised manuscript.

Another element that would give a good indication of the ability of THAV to infect neural cells would be to document the ability to achieve in vitro infections using a human cell line of neural/neuronal origin, such as SHSY5Y or U-373MG cells (non-limitative list).

As mentioned above, we have now included data of the infection of LUHMES cells, a human-derived cell line that can be differentiated in post-mitotic neurons. We confirmed by RT-qPCR and smFISH that these cells were readily infected by TAHV, highlighting that human neurons are permissive to TAHV infection. These data have been added in new **Figure 4E-F** of the revised manuscript.

6. Although it discussed later in the manuscript, the rationale for testing the antiviral potential of RG10b is not explained with sufficient clarity. The authors indicate that they previously demonstrated the efficacy of this molecule on Coronaviruses, but this may not be a very valid argument. Indeed, CoVs are positive stranded RNA viruses, whereas THAV is a segmented negative-stranded RNA. Rather, the authors could indicate that the putative mechanisms of action of RG10b (which is still not fully understood, but may be linked to perturbations at the level of the ER) may be applicable to bunyaviruses.

As suggested by the reviewer, we included some more info to explain the rationale of using RG10b against TAHV in the Discussion section.

Other points:

1. The authors need to be consistent and more careful about their use of tenses. In particular, the results and methods section should be written in the past tense throughout.

This has been corrected

2. The authors should indicate the agreement numbers issued by the agencies that approved the study.

This has been added

3. "media" should be replaced by "medium" (see for instance lines 366 to 376, or 430 to 437).

This has been changed

4. I am not sure that it is really necessary to abbreviate "air-liquid interface" by ALI

The abbreviation has been removed.

Referee #3 (Comments on Novelty/Model System for Author):

The concept taken by authors are novel in the field of virology and its related domains.

We thank the reviewer for acknowledging the novelty of our work.

Referee #3 (Remarks for Author):

The manuscript is well conceptualize and novel in the field of virology. However there are three donors were included in this study. Data variation among these three donors are prominent which makes the hypothesis weak. Therefore, adding more donors in experimental setup may enhance the power of study and relevance of hypothesis strongly.

We thank the reviewer for his/her comment. Although our plan is indeed to screen more compounds using many more donors in the coming years, we weren't able to do so during the revision process, but we believe that our study still provides very important information to the field. Indeed, data variation is particularly interesting, and we do not think it makes our hypothesis weak, but rather perfectly exemplify that a drug that works very well in cell lines, may not show as much potency in more physiological context such as OPAB. It is often wondered why small molecules are very efficient in vitro, or even in animal models, but fail in clinical trials. The OPAB model could actually be a critical preclinical tool to evaluate drug candidates in greater details. With these three donors, we illustrate this concept, hoping to drive the future of antiviral research toward more reliable cellular models.

25th Jan 2024

Dear Dr. Gaudin,

Thank you for the submission of your revised manuscript to EMBO Molecular Medicine. Please find enclosed the final reports on your manuscript. I am pleased to inform you that we will be able to accept your manuscript pending the following final amendments:

- 1) Authors: There is a discrepancy between Stéphan Mora in our system and Stephan Mora in the manuscript. Please correct.
- 2) Source data: Please submit completed source data checklist.
- 3) In the main manuscript file, please do the following:
 - Please address all comments suggested by our data editors listed below:
 - o Figure legends:
 1. Please note that a separate 'Data Information' section is required in the legends of figures EV 3a-c, e.
 2. Please note that information related to n is missing in the legends of figures 5c; 6a; EV 2a-b, d-e; EV 3d; EV 4b.
 3. Please note that the error bars are not defined in the legends of figures 2f-h; 5c; 6a; EV 2a-b, d-e; EV 3d; EV 4b.
 4. Please note that the scale bar needs to be defined for figures 1b-e; 2a-d; 3a-c; 5e; EV 2c, f.
 - Reduce keywords to max. 5.
 - In M&M, provide the antibody dilutions that were used for each antibody.
 - In M&M, provide the statement that informed consent was obtained from all human subjects and confirm that the experiments conformed to the principles set out in the WMA Declaration of Helsinki and the Department of Health and Human Services Belmont Report.
 - In M&M, add statistical paragraph that should reflect all information that you have filled in the Authors Checklist, especially regarding randomization, blinding, replication.
 - Place "Data availability" at the end of M&M section and remove the sentence "Other data are available in the text or upon reasonable request."
 - Author contributions: Please remove it from the manuscript and specify author contributions in our submission system. CRediT has replaced the traditional author contributions section because it offers a systematic machine-readable author contributions format that allows for more effective research assessment. You are encouraged to use the free text boxes beneath each contributing author's name to add specific details on the author's contribution. More information is available in our guide to authors:
<https://www.embopress.org/page/journal/17574684/authorguide#authorshipguidelines>
 - Correct the reference citation in the text and reference list. In the text a reference should be cited by author and year of publication. Include a space between a word and the opening parenthesis of the reference that follows. In the reference list, citations should be listed in alphabetical order. Where there are more than 10 authors on a paper, 10 will be listed, followed by "et al.". Also, please remove DOIs. Please check "Author Guidelines" for more information.
<https://www.embopress.org/page/journal/17574684/authorguide#referencesformat>
- 4) Tables and Figures: Please rename Table EV2 to Dataset EV1 and update tables' callouts in the main manuscript text to Table EV1 and Dataset EV1. Also, remove the table legends from the main manuscript file and add them to the table excel files. Expanded View Figures should be renamed to Figure EV1 etc.
- 5) The Paper Explained: Please add it to the main manuscript file.
- 6) Synopsis:
 - Please resize synopsis image to 550 px-wide x (250-400)-px high and upload it as a high-resolution .jpeg file.
 - Please check your synopsis text and image before submission with your revised manuscript. Please be aware that in the proof stage minor corrections only are allowed (e.g., typos).
- 7) For more information: This space should be used to list relevant web links for further consultation by our readers. Could you identify some relevant ones and provide such information as well? Some examples are patient associations, relevant databases, OMIM/proteins/genes links, author's websites, etc...
- 8) As part of the EMBO Publications transparent editorial process initiative (see our Editorial at <http://embomolmed.embopress.org/content/2/9/329>), EMBO Molecular Medicine will publish online a Review Process File (RPF) to accompany accepted manuscripts. This file will be published in conjunction with your paper and will include the anonymous referee reports, your point-by-point response and all pertinent correspondence relating to the manuscript. Let us know whether you agree with the publication of the RPF and as here, if you want to remove or not any figures from it prior to publication. Please note that the Authors checklist will be published at the end of the RPF.
- 9) Please provide a point-by-point letter INCLUDING my comments as well as the reviewer's reports and your detailed responses (as Word file).

I look forward to reading a new revised version of your manuscript as soon as possible.

Yours sincerely,

Zeljko Durdevic

*** Instructions to submit your revised manuscript ***

- 1) a .docx formatted version of the manuscript text (including Figure legends and tables)
- 2) Separate figure files*
- 3) supplemental information as Expanded View and/or Appendix. Please carefully check the authors guidelines for formatting Expanded view and Appendix figures and tables at <https://www.embopress.org/page/journal/17574684/authorguide#expandedview>
- 4) a letter INCLUDING the reviewer's reports and your detailed responses to their comments (as Word file).
- 5) The paper explained: EMBO Molecular Medicine articles are accompanied by a summary of the articles to emphasize the major findings in the paper and their medical implications for the non-specialist reader. Please provide a draft summary of your article highlighting
 - the medical issue you are addressing,
 - the results obtained and
 - their clinical impact.This may be edited to ensure that readers understand the significance and context of the research. Please refer to any of our published articles for an example.
- 6) For more information: There is space at the end of each article to list relevant web links for further consultation by our readers. Could you identify some relevant ones and provide such information as well? Some examples are patient associations, relevant databases, OMIM/proteins/genes links, author's websites, etc...
- 7) Author contributions: the contribution of every author must be detailed in a separate section.
- 8) EMBO Molecular Medicine now requires a complete author checklist (<https://www.embopress.org/page/journal/17574684/authorguide>) to be submitted with all revised manuscripts. Please use the checklist as guideline for the sort of information we need WITHIN the manuscript. The checklist should only be filled with page numbers where the information can be found. This is particularly important for animal reporting, antibody dilutions (missing) and exact values and n that should be indicated instead of a range.
- 9) Every published paper now includes a 'Synopsis' to further enhance discoverability. Synopses are displayed on the journal webpage and are freely accessible to all readers. They include a short stand first (maximum of 300 characters, including space) as well as 2-5 one sentence bullet points that summarise the paper. Please write the bullet points to summarise the key NEW findings. They should be designed to be complementary to the abstract - i.e. not repeat the same text. We encourage inclusion of key acronyms and quantitative information (maximum of 30 words / bullet point). Please use the passive voice. Please attach

these in a separate file or send them by email, we will incorporate them accordingly.

You are also welcome to suggest a striking image or visual abstract to illustrate your article. If you do please provide a jpeg file 550 px-wide x 300-800px high.

10) A Conflict of Interest statement should be provided in the main text

11) Please note that we now mandate that all corresponding authors list an ORCID digital identifier. This takes <90 seconds to complete. We encourage all authors to supply an ORCID identifier, which will be linked to their name for unambiguous name identification.

Currently, our records indicate that the ORCID for your account is 0000-0002-8558-8143.

Link Not Available

Photos 400-800 DPI

*Additional important information regarding figures and illustrations can be found at

<https://bit.ly/EMBOPressFigurePreparationGuideline>. See also figure legend preparation guidelines:

<https://www.embopress.org/page/journal/17574684/authorguide#figureformat>

***** Reviewer's comments *****

Referee #1 (Comments on Novelty/Model System for Author):

Adequately developed and described.

Referee #1 (Remarks for Author):

I recommend the manuscript for publication in its current form.

Referee #2 (Comments on Novelty/Model System for Author):

The authors have provided convincing evidence that their model system is truly adequate for their intended use. My initial doubts are totally raised

Referee #2 (Remarks for Author):

The authors have to be congratulated for their efforts and the work undertaken to address all issues that were raised in my first review. I am now entirely convinced by the importance and interest of this work, as all challenging points have been addressed very satisfactorily.

The authors addressed the minor editorial issues.

5th Feb 2024

Dear Dr. Gaudin,

We are pleased to inform you that your manuscript is accepted for publication and is now being sent to our publisher to be included in the next available issue of EMBO Molecular Medicine.
